# A Euclidean transformer for fast and stable machine learned force fields

J. Thorben Frank[1,2], Oliver T. Unke[3], Klaus-Robert Müller ⓘ [1,2,3,4,5] ✉ & Stefan Chmiela ⓘ [1,2] ✉

Recent years have seen vast progress in the development of machine learned force fields (MLFFs) based on ab-initio reference calculations. Despite achieving low test errors, the reliability of MLFFs in molecular dynamics (MD) simulations is facing growing scrutiny due to concerns about instability over extended simulation timescales. Our findings suggest a potential connection between robustness to cumulative inaccuracies and the use of equivariant representations in MLFFs, but the computational cost associated with these representations can limit this advantage in practice. To address this, we propose a transformer architecture called SO3KRATES that combines sparse equivariant representations (*Euclidean variables*) with a self-attention mechanism that separates invariant and equivariant information, eliminating the need for expensive tensor products. SO3KRATES achieves a unique combination of accuracy, stability, and speed that enables insightful analysis of quantum properties of matter on extended time and system size scales. To showcase this capability, we generate stable MD trajectories for flexible peptides and supra-molecular structures with hundreds of atoms. Furthermore, we investigate the PES topology for medium-sized chainlike molecules (e.g., small peptides) by exploring thousands of minima. Remarkably, SO3KRATES demonstrates the ability to strike a balance between the conflicting demands of stability and the emergence of new minimum-energy conformations beyond the training data, which is crucial for realistic exploration tasks in the field of biochemistry.

Atomistic modeling relies on long-timescale molecular dynamics (MD) simulations to reveal how experimentally observed macroscopic properties of a system emerge from interactions on the microscopic scale[1]. The predictive accuracy of such simulations is determined by the accuracy of the interatomic forces that drive them. Traditionally, these forces are either obtained from exceedingly approximate mechanistic force fields (FF) or accurate, but computationally prohibitive ab initio electronic structure calculations. Recently, machine learning (ML) potentials have started to bridge this gap, by exploiting statistical dependencies of molecular systems with so far unprecedented flexibility[2–27].

The accuracy of MLFFs is traditionally determined by their test errors on a few established benchmark datasets[8,28,29]. Despite providing an initial estimate of MLFF accuracy, recent research[30–33] indicates that there is only a weak correlation between MLFF test errors and their performance in long-timescale MD simulations, which is considered the true measure of predictive usefulness. Faithful representations of dynamical and thermodynamic observables can only be derived from accurate MD trajectories. From an ML perspective this shortcoming can be attributed to poor extrapolation behavior, which becomes particularly severe for high-temperature configurations or conformationally flexible structures. In these cases, the geometries

[1]Machine Learning Group, TU Berlin, Berlin, Germany. [2]BIFOLD, Berlin Institute for the Foundations of Learning and Data, Berlin, Germany. [3]Google DeepMind, Berlin, Germany. [4]Department of Artificial Intelligence, Korea University, Seoul, Korea. [5]Max Planck Institut für Informatik, Saarbrücken, Germany. ✉e-mail: klaus-robert.mueller@tu-berlin.de; stefan@chmiela.com

explored during MD simulations significantly deviate from the distribution of the training data.

The ongoing progress in MLFF development has resulted in a wide range of increasingly sophisticated model architectures aiming to improve the extrapolation behavior. Among these, message passing neural networks (MPNNs)[9,12,34] have emerged as a particularly effective class of architectures. MPNNs can be considered as a generalization of convolutions to handle unstructured data domains, such as molecular graphs. This operation provides an effective way to extract features from the input data and is ubiquitous in many modern ML architectures. Recent advances in this area focused on the incorporation of physically meaningful geometric priors[8,11,23,35,36]. This has lead to so-called *equivariant* MPNNs, which have been found to reduce the obtained approximation error[36–40] and offer better data efficiencies than invariant models[36]. Invariant models rely on pairwise distances to describe atomic interactions, as they do not change upon rotation[5]. However, with growing system size, flexibility or chemical heterogeneity, it becomes increasingly harder to derive the correct interaction patterns within this limited representation. This is why equivariant models enable to incorporate additional directional information, to capture interactions depending on the relative orientation of neighboring atoms. It allows them to discriminate interactions that can appear inseparable to simpler models[37,41] and to learn more transferable interaction patterns from the same training data.

A fundamental building block of most equivariant architectures is the tensor product. It is evaluated within the convolution operation $(f * g)(x)$ between pairs of functions $f(x)$ and $g(x)$ expanded in linear bases[42]. The result is then defined in the product space of the original basis function sets. Thus, the associated product space quickly becomes computationally intractable as it grows exponentially in the number of convolution operations.

In SO(3) equivariant architectures, convolutions are performed over the SO(3) group of rotations in the basis of the *spherical harmonics*. The exponential growth of the associated function space is avoided by fixing the maximum degree $l_{max}$ of the spherical harmonics in the architecture. The maximum degree has been shown to be closely connected to accuracy, data efficiency[24,36] and linked to the reliability of the model in MD simulations. However, SO(3) convolutions scale as $l_{max}^6$, which can increase the prediction time per conformation by up to two orders of magnitude compared to an invariant model[32,43]. This has lead to a situation where one has to compromise between accuracy, stability and speed, which can pose significant practical problems that need to be addressed before such models can become useful in practice for high-throughput or extensive exploration tasks.

We take this as motivation to propose a *Euclidean self-attention* mechanism that replaces SO(3) convolutions with a filter on the relative orientation of atomic neighborhoods, representing atomic interactions without the need for expensive tensor products. Our solution builds on recent advances in neural network architecture design[44] and from the field of geometric deep learning[36,37,39,45]. Our SO3KRATES method uses a sparse representation for the molecular geometry and restricts projections of all convolution responses to the most relevant invariant component of the equivariant basis functions. Due to the orthonormality of the spherical harmonics, such a projection corresponds to the trace of the product-tensor, which can be expressed in terms of a linear-scaling inner product. This enables efficient scaling to high-degree equivariant representations without sacrificing computational speed and memory cost. Force predictions are obtained from the gradient of the resulting invariant energy model, which represents a piece-wise linearization that is naturally equivariant. Throughout, a self-attention mechanism is used to decouple invariant and equivariant basis elements within the model. We compare the stability and speed of the proposed SO3KRATES model with current state-of-the art ML potentials and find that our solution overcomes the limitations of current

equivariant MLFFs, without compromising on their advantages. Our proposed mathematical formulation leading to an efficient equivariant architecture enables reliably stable MD simulations with a speedup of up to a factor of ~30 over equivariant MPNNs with comparable stability and accuracy[32]. To demonstrate this, we run accurate nanosecond-long MD simulations for supra-molecular structures within only a few hours, which allows us to calculate Fourier transforms of converged velocity auto-correlation functions for structures that range from small peptides with 42 atoms up to nanostructures with 370 atoms. We further apply our model to explore the topology of the PES of docosahexaenoic acid (DHA) and Ac-Ala3-NHMe by investigating 10k minima using a minima hopping algorithm[46]. Such an investigation requires roughly 30M FF evaluations that are queried at temperatures between a few 100 K up to ~1200 K. With DFT methods, this analysis would require more than a year of computation time. Existing equivariant MLFFs with comparable prediction accuracy would run more than a month for such an analysis. In contrast, we are able to perform the simulation in only 2.5 days, opening up the possibility to explore hundreds of thousands of PES minima on practical timescales. Furthermore, we show that SO3KRATES enables the detection of physically-valid minima conformations which have not been part of the training data. The ability to extrapolate to unknown parts of the PES is essential for scaling MLFFs to large structures, since the availability of ab-initio reference data can only cover sub-regions for conformationally rich structures.

We also examine the impact of disabling the equivariance property in our network architecture to gain a deeper understanding of its influence on the characteristics of the model and its reliability in MD simulations. Here we find, that the equivariant nature can be linked to the stability of the resulting MD simulation and to the extrapolation behavior to higher temperatures. We are able to show, that equivariance lowers the spread in the error distribution even when the test error estimate is the same on average. Thus, using directional information via equivariant representations shows analogies in spirit to classical ML theory, where mapping into higher dimensions yields richer features spaces that are easier to parametrize[47–49].

## Results

### From equivariant message passing neural networks to separating invariant and equivariant structure: SO3KRATES

MPNNs[34] carry over many of the properties of convolutions to unstructured input domains, such as sets of atomic positions in Euclidean space. This has made them one promising approach for the description of the PES[12,17,24,36,50–52], where the potential energy is typically predicted as

$$E_{pot}(\vec{r}_1, \dots, \vec{r}_n) = \sum_{i=1}^{n} E_i. \tag{1}$$

The energy contributions $E_i \in \mathbb{R}$ are calculated from high-dimensional atomic representations $f_i^{[T]}$. They are constructed iteratively (from $T$ steps), by aggregating pairwise messages $m_{ij}$ over atomic neighborhoods $\mathcal{N}(i)$

$$f_i^{[t+1]} = \text{UPD}\left( f_i^{[t]}, \bigoplus_{j \in \mathcal{N}(i)} m_{ij} \right), \tag{2}$$

where UPD( · ) is an update function that mixes the representations from the prior iteration and the aggregated messages.

One way of incorporating the rotational invariance of the PES is to build messages that are based on potentially incomplete sets of invariant inputs such as distances, angles or dihedral angles. An alternative is to use SO(3) equivariant representations[36,38,39,52] within a

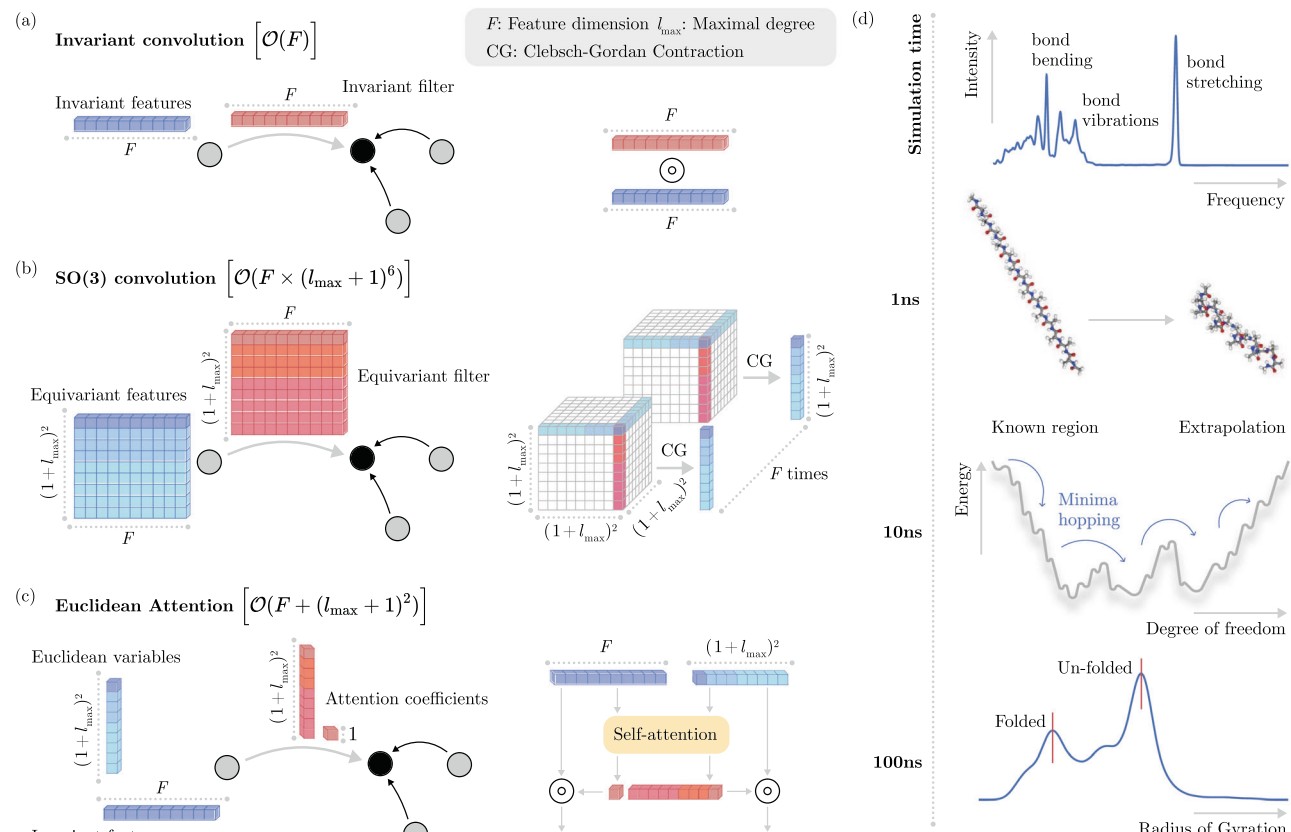

**Fig. 1 | Results overview. a** Illustration of an invariant convolution. **b** Illustration of an SO(3) convolution. **c** Illustration of the Euclidean attention mechanism that underlies the SO3KRATES transformer. We decompose the representation of molecular structure into high dimensional invariant features and equivariant Euclidean variables (EV), which interact via self-attention. **d** The combination of simulation stability and computational efficiency of SO3KRATES enables the analysis of a broad set of properties (power spectra, folding dynamics, minima analysis, radius of gyration) on different simulation timescales.

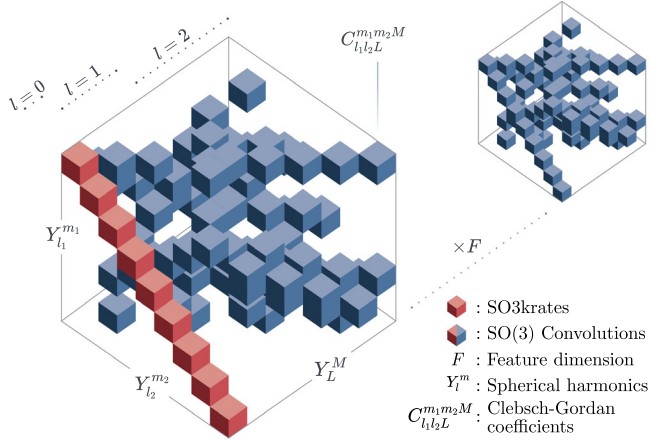

**Fig. 2 | Learning on invariants.** SO(3) convolutions are constructed as triplet tensor products in the spherical harmonics basis, which is performed $F$ times along the feature dimension $F$. We replace SO(3) convolutions by a parametrized filter function on the invariants (red blocks), which effectively reduces the tripled tensor product to taking the partial (per-degree) trace of a simple tensor product. Colored volumes correspond to the non-zero entries in the Clebsch-Gordan coefficients $C_{l_1 l_2 L}^{m_1 m_2 M}$, which mask the tensor products, $l$ denotes the degree and $Y_l^m$ is the spherical harmonics function.

basis that allows for systematic multipole expansion of the geometry to match the complexity of the modeled system.

This requires to generalize the concept of invariant continuous convolutions[12] to the SO(3)-group of rotations. A message function

performing an SO(3) convolution can be written as[36,42]

$$m_{ij}^{LM} = \sum_{l_1 l_2 m_1 m_2} C_{l_1 l_2 L}^{m_1 m_2 M} \phi^{l_1 l_2 L}(r_{ij}) Y_{l_1}^{m_1}(\hat{r}_{ij}) f_j^{l_2 m_2}, \quad (3)$$

where $C_{l_1 l_2 L}^{m_1 m_2 M}$ are the *Clebsch-Gordan coefficients*, $Y_l^m$ is a spherical harmonic of degree $l$ and order $m$, the function $\phi^{l_1 l_2}: \mathbb{R} \mapsto \mathbb{R}^F$ modulates the radial part and $f_j^{l_2 m_2} \in \mathbb{R}^F$ is an atomic feature vector. Thus, performing a single convolution scales as $\mathcal{O}(l_{max}^6 \times F)$, where $l_{max}$ is the largest degree in the network Fig. 1b and Fig. 2). Here we *propose* two conceptual changes to Eq. (3) that we will denote as Euclidean self-attention: (1) We separate the message into an invariant and an equivariant part and (2) replace the SO(3) convolution by an attention function on its invariant output. To do so, we start by initializing atomic features $f_i^{[t=0]} \in \mathbb{R}^F$ and *Euclidean variables* (EV) $x_{i,LM}^{[t=0]} \in \mathbb{R}$ from the atomic types and the atomic neighborhoods, respectively. Collecting all orders and degrees for the EV in a single vector, gives $(l_{max}+1)^2$ dimensional representations $\mathbf{x}_i$ that transform equivariant under rotation and $l_{max}$ ("Methods" section IV A).

1.  The message for the invariant components is expressed as

$$m_{ij} = \alpha_{ij} f_j, \quad (4)$$

whereas the equivariant parts propagate as

$$m_{ijLM} = \alpha_{ij,L} Y_L^M(\hat{r}_{ij}), \quad (5)$$

**Table 1 | Computational complexity**

| Architecture | Scaling | $l_{max}$ |
|---|---|---|
| SCHNET[9] | $\mathcal{O}(n \times \langle \mathcal{N} \rangle \times 1 \times F)$ | 0 |
| PAINN[37] | $\mathcal{O}(n \times \langle \mathcal{N} \rangle \times 4 \times F)$ | 1 |
| SPOOKYNET[24] | $\mathcal{O}(n \times \langle \mathcal{N} \rangle \times (l_{max}+1)^2 \times F)$ | 2 |
| NEQUIP[36] | $\mathcal{O}(n \times \langle \mathcal{N} \rangle \times (l_{max}+1)^6 \times F)$ | 3 |
| SO3KRATES | $\mathcal{O}(n \times \langle \mathcal{N} \rangle \times ((l_{max}+1)^2 + F))$ | 3 |

Scaling for different message passing architectures, where $n$ is the number of atoms, $\langle \mathcal{N} \rangle$ the average number of neighbors, $l_{max}$ the maximal degree and $F$ the feature dimension. SCHNET and PAINN have fixed maximal degree of $l_{max}=0$ and $l_{max}=1$ whereas they are free parameter in other models.

where $\alpha_{ij} \in \mathbb{R}$ are (per-degree) *attention coefficients*. Features and EV are updated with the aggregated messages to

$$f_i^{[t+1]} = f_i^{[t]} + \sum_{j \in \mathcal{N}(i)} m_{ij}, \quad (6)$$

for the features and

$$x_{iLM}^{[t+1]} = x_{iLM}^{[t]} + \sum_{j \in \mathcal{N}(i)} m_{ijLM}, \quad (7)$$

respectively. Due to this separation, the overall message calculation scales as $\mathcal{O}(l_{max}^2 + F)$, as it replaces the multiplication of feature dimension and $l_{max}$ that appears in other equivariant architectures by an addition (Table 1). As shown in[53], a separation between invariant and equivariant representations can also achieved by adding an invariant latent space that is updated using iterated tensor products on an equivariant, edge based feature space. In contrast, our approach is centered around atom-wise representations and the a priori separation of both interaction spaces allows to fully avoid the usage of tensor products. Both design choices benefit computational efficiency.

2. Instead of performing full SO(3) convolutions, we move the learning of complex interaction patterns into an attention function

$$\alpha_{ij} = \alpha \left( f_i, f_j, r_{ij}, \oplus_{l=0}^{l_{max}} \boldsymbol{x}_{ij, l \to 0} \right), \quad (8)$$

where $\oplus_{l=0}^{l_{max}} \boldsymbol{x}_{ij, l \to 0}$ is the invariant output of the SO(3) convolution over the EV signals located on atom $i$ and $j$ ("Methods" section IV B). Thus, Eq. (8) non-linearly incorporates information about the relative orientation of atomic neighborhoods. Since the Clebsch-Gordan coefficients are diagonal matrices along the $l=0$ axis (Fig. 2), calculating the invariant projections requires to take per-degree traces of length $(2l+1)$ and can be computed efficiently in $\mathcal{O}(l_{max}^2)$. Within SO3KRATES atomic representations are refined iteratively as

$$[\mathbf{f}_i^{[t+1]}, \boldsymbol{x}_i^{[t+1]}] = \text{ECTBLOCK} \left[ \{\mathbf{f}_j^{[t]}, \boldsymbol{x}_j^{[t]}, \vec{r}_{ij}\}_{j \in \mathcal{N}(i)} \right], \quad (9)$$

where each Euclidean transformer block (ECTBLOCK) consists of a self-attention block and an interaction block. The self-attention block, implements the Euclidean self-attention mechanism described in the previous section. The interaction block gives additional freedom for parametrization by exchanging information between features and EV located at the same atom. After $T$ MP steps, per-atom energies $E_i$ are calculated from the final features $f_i^{[T]}$ using a two-layered neural network and are summed to the total potential energy (Eq. (1)). Atomic forces are obtained using automatic differentiation, which ensures energy conservation[8].

We remark that the outlined equivariant architecture does not preclude the modeling of vectorial and tensorial properties, such as atomic quadrupoles or octopoles, up to the set maximum degree $l_{max}$. For example, molecular dipoles can be learned by combining invariant partial charge predictions with atomic dipoles extracted from the EVs of degree $l=1$[24,37].

A detailed outline of the architectural components and the proposed Euclidean self-attention framework is given in the "Methods" section and in Fig. 3.

**Overcoming accuracy-stability-speed trade-offs**

We now demonstrate in the following numerical experiment, that SO3KRATES can overcome the trade-offs between MD stability, accuracy and computational efficiency (Fig. 4).

A recent study compared the stability of different state-of-the-art MLFFs in short MD simulations and found that only the SO(3) convolution-based architecture NEQUIP[36] gave reliable results[32]. However, the excellent stability of such models comes at a substantial computational cost associated with this operation (Fig. 4a, top panel). This necessitates a trade-off between the stability and the computational efficiency of the MLFF, which SO3KRATES can now overcome (Fig. 4b). Our model allows the prediction of up to one order of magnitude more frames per second (FPS) (Fig. 4c), enabling step times at sub-millisecond speed, without sacrificing reliability or accuracy in MD simulations (Fig. 4 (b)). We remark however, that test accuracy and stability do not necessarily correlate (compare e.g., GEMNET and SPHERENET in Fig. 4a) but only simultaneously accurate *and* stable models are of practical interest. We find, that SO3KRATES yields accurate force predictions, thus overcoming this trade-off effectively (Fig. 4b). As for stability and speed, the investigated models in ref. 32 show an accuracy-speed trade-off (Fig. 4a, lower panel) in line with the findings reported in ref. 54.

Any empirical runtime measurement depends on specific hardware and software conditions. The run times reported in[32] have been measured for MLFF models implemented in PYTORCH+ASE. To ensure comparability with SO3KRATES (implemented in JAX), we re-implement two representative models along the trade-off lines (Fig. 4a) under SO3KRATES-identical settings. Since the study mentioned above reports that the fastest model (FORCENET) yields wrong observables in their benchmark, we instead chose the second fastest contender (SCHNET) for our re-implementation. As the most stable and accurate model we chose NEQUIP for re-implementation. This selection of architectures is also representative for invariant (SCHNET) and equivariant SO(3)-convolution-based models (NEQUIP), constituting the upper and lower bounds in terms of computational complexity (Table 1). All models are re-implemented in JAX[55] using the E3X library[56]. MD step times are measured with the same MD code written in JAX-MD[57] on the same physical device (Nvidia V100 GPU). The models follow the default MD17 hyperparameters as outlined in the original publications[12,36]. This ensures equal footing for our runtime comparisons that follow. Interestingly, the transition from PYTORCH+ASE (Fig. 4b) purple dashed line) to JAX+JAX-MD (Fig. 4b) purple solid line) allows for a speed-up between 28 for NEQUIP and 15 for SCHNET. This illustrates the importance of identical settings and the potential of the JAX ecosystem. Notably, the step times are measured without the time required for IO operations, since they are highly dependent on the local HPC infrastructure. Thus, wall times we report for full simulations have a constant offset w. r. t. the reported step times.

For small organic molecules with up to 21 atoms SO3KRATES achieves an averaged speed-up by a factor of 5 compared to the NEQUIP architecture whereas step times are slightly larger (by a factor of 1.4) than for the invariant SCHNET model. The speed-up over SO(3) convolutions increases in the total number of atoms (Fig. 4c), which is in line with the smaller pre-factor in the theoretical scaling analysis (Table 1) such that for the double walled nanotube (370 atoms), the speed-up compared to NEQUIP has grown to a factor of 30. Compared to invariant convolutions, we find our approach to yield a slightly slower prediction speed which is in line with theoretical considerations.

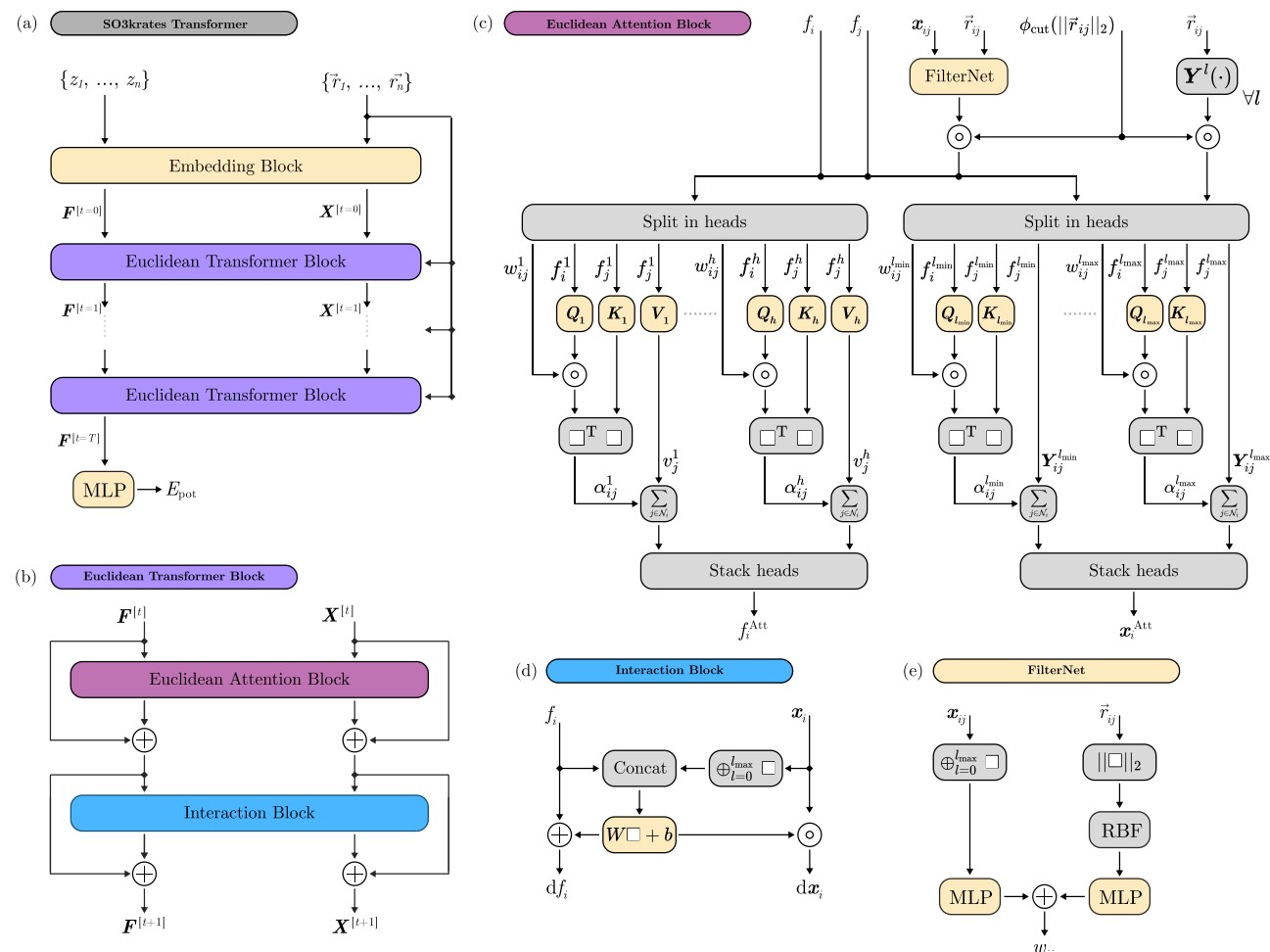

**Fig. 3 | SO3krates architecture and building blocks. a** The SO3krates transformer takes atomic types $Z = \{z_1, ..., z_n\}$ and atomic positions $R = \{\vec{r}_1, ..., \vec{r}_n\}$ and returns a corresponding energy $E_{pot}$. Within the embedding block $Z$ and $R$ are embedded into invariant features $F$ and equivariant Euclidean variables (EV) $X$. They are refined via $T$ Euclidean transformer blocks. Per-atom energies are predicted from the final features via a multilayer perceptron (MLP) and are summed up to yield the total energy. **b** A Euclidean transformer block consists of a Euclidean attention block and an interaction block. Both blocks are enveloped by skip connections which allow to carry over information from prior layers. **c** The Euclidean attention block aggregates atomic neighborhood information. It consists of two branches one updating the features and one the EV. Within the feature branch the invariant features and the filter vectors $w_{ij}$ are split into $h$ attention heads. Query, key and value vector are constructed from the features via trainable matrices $Q$, $K$ and $V$. Query and filter are multiplied point-wise before a dot product with the key vector yields attention

coefficients $\alpha_{ij}$ which weight the value vectors during neighborhood aggregation. The aggregated per-head features are stacked back together and yield a single attended feature $f_i^{Att}$. The EV branch follows a similar design with the number of heads being determined by the minimum ($l_{min}$) and maximal degree ($l_{max}$) in the network. Instead of an invariant value vector, spherical harmonics vectors $Y_{ij}^l$ from minimal to maximal degree are used for the different heads. **d** The interaction block exchanges per-atom information between features and EV. It contracts the EV to per-degree invariants and concatenates the result with the invariant features. The concatenated result is passed through an affine transformation which gives updates for the features and the EV. **e** A pairwise difference vector between EV is contracted to per-degree invariants and passed through an MLP. From the atomic displacement vector the pairwise distance is calculated and expanded in radial basis functions before fed into an MLP. The MLP outputs are summed to yield the filter vector $w_{ij}$.

For the radial distribution functions (RDFs), we find consistent results across five simulation runs (Supplementary Fig. 3) for all of the four investigated structures, which are in agreement with the RDFs from DFT calculations. Interestingly, it has been found that some faster MLFF models can give inaccurate RDFs, which result in MAEs between 0.35 for salicylic acid and 1.02 for naphthalene[32]. In comparison the achieved accuracies with SO3krates show that the seemingly contradictory requirements of high computational speed and accurate observables from MD trajectories can be reconciled.

A recent work, proposed a strictly local equivariant architecture, called ALLEGRO[53]. This allows for parallelization without additional communication, whereas parallelization of MPNNs with $T$ layers requires $T-1$ additional communication calls between computational nodes. On the example of the $Li_3PO_4$ solid electrolyte we compare accuracy and speed to the ALLEGRO model for a unit cell with 192 atoms (Table 2). For the recommended hyperparameter settings, SO3krates

achieves energy and force accuracies, more than 50% better than the ones reported in ref. 53, even with only one tenth of the training data. At the same time, the timings in MD simulations are on par. Notably the model settings in ref. 53 have been optimized for speed rather than accuracy in order to demonstrate scalability. This again expresses an accuracy-speed trade-off that we can improve upon using the SO3krates architecture. To further validate the physical correctness of the obtained MD trajectory, we compare the RDFs at 600K to the ones obtained from DFT in the quenched phase of $Li_3PO_4$ (Supplementary Fig. 7). The results showcase the applicability of SO3krates to materials, beyond molecular structures.

## Data efficiency, stability and extrapolation

Data efficiency and MD stability play an important role for the applicability of a MLFFs. High data efficiency allows to obtain accurate PES approximations even when only little data is available, which is a

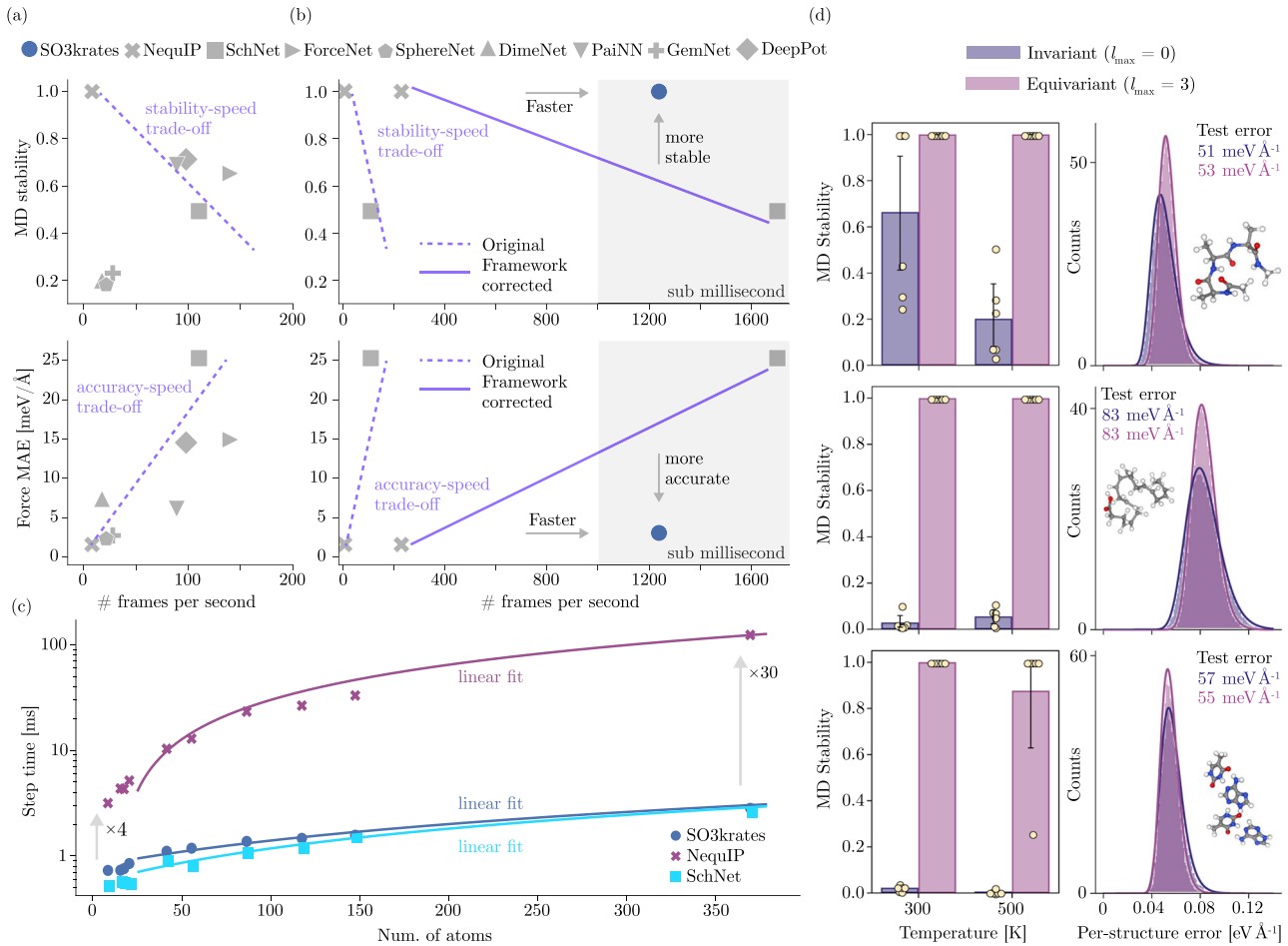

**Fig. 4 | Computational efficiency and molecular dynamics (MD) stability.**
**a** Number of frames per second (FPS) vs. the averaged stability coefficient (upper panel) and FPS vs. the averaged force mean absolute error (MAE) (lower panel) for four small organic molecules from the MD17 data set as reported in[32] for different state-of-the-art MPNN architectures[12,36,37,43,50,94–96]. **b** Since run times are sensitive to hardware and software implementation details, we re-implement two representative models along the trade-off lines under settings identical to the SO3KRATES MLFF (using JAX), which yields framework-corrected FPS (dashed vs. solid line). We observe speed-ups between 28 (for NEQUIP) and 15 (for SCHNET) in our re-implementations. We find, that SO3KRATES enables reliable MD simulations and high accuracies without sacrificing computational performance. Gray shaded area indicates the regime of sub-millisecond step time. **c** MD step time vs. the number of

atoms in the system. The smaller pre-factor in the computational complexity compared to SO(3) convolutions (Table 1) results in computational speed-ups that grow in system size. **d** MD stability observed at temperatures 300 K and 500 K. The transition to higher temperatures results in a drop of stability for the invariant model, hinting towards less robustness and weaker extrapolation behavior. Flexible molecules such as DHA pose a challenge for the invariant model at 300 K already. Bar height is the mean stability over six MD runs and yellow dots denote stability for individual MD runs. Error bars correspond to the $2\sigma$ confidence interval. Per-structure error distributions for an invariant and an equivariant SO3KRATES model with the same error on the test set. Spread and mean of the error distributions are given in Supplementary Table 1.

common setting due to the computational complexity of quantum mechanical ab-initio methods. Even when high accuracies can be achieved, without MD stability the calculation of physical observables from the trajectories becomes impossible. Here, we show that the data efficiency of SO3KRATES can be successively increased further by increasing the maximum degree $l_{max}$ in the network (Supplementary

Fig. 4). We further find, that the stability and extrapolation to higher temperatures of the MLFF can be linked to the presence of equivariant representations, independent of the test error estimate (Fig. 4 (d)).

To understand the benefits of directional information, we use an equivariant ($l_{max} = 3$) and an invariant model ($l_{max} = 0$) within our analysis. Due to the use of multi-head attention, the change in the number of network parameters is negligible when going from $l_{max} = 0$ to $l_{max} = 3$ ("Methods" section IV H). All models were trained on 11k randomly sampled geometries from which 1k are used for validation. This number of training samples was necessary to attain force errors close to 1 kcal mol$^{-1}$ Å$^{-1}$ for the invariant model. Since equivariant representations increase the data efficiency of ML potentials[24,36], we expect the equivariant model to have a smaller test error estimate given the same number of training samples. We confirm this expectation on the example of the DHA molecule, where we compare the data efficiency for different degrees $l_{max}$ on the example of the DHA molecule (Supplementary Fig. 4). To make the comparison of invariant and equivariant model as fair as possible, we train the invariant model

**Table 2 | Accuracy and speed for periodic systems**

| | $n_{train}$ | $E_{MAE}$ [$\frac{meV}{atom}$] | $F_{MAE}$ [$\frac{meV}{Å}$] | $\frac{\mu s}{step\,atom}$ |
|---|---|---|---|---|
| ALLEGRO[53] | 10k | 1.7 | 73.4 | 27.785* |
| SO3KRATES | 10k | 0.2 | 28.2 | 23.593* |
| SO3KRATES | 1k | 0.3 | 31.8 | 23.593* |

Speed in MD simulation and accuracy comparison to the strictly local ALLEGRO model for Li$_3$PO$_4$ (192 atoms) on a single V100 GPU as reported in ref. 53. In ref. 53 no inference times and only MD step times with LAMMPS[93] have been reported. This prohibits a purely model based comparison. We run MD simulations using the MDX code[92], such that timings should be understood as illustration of the competitive nature in speed rather than an exact comparison.

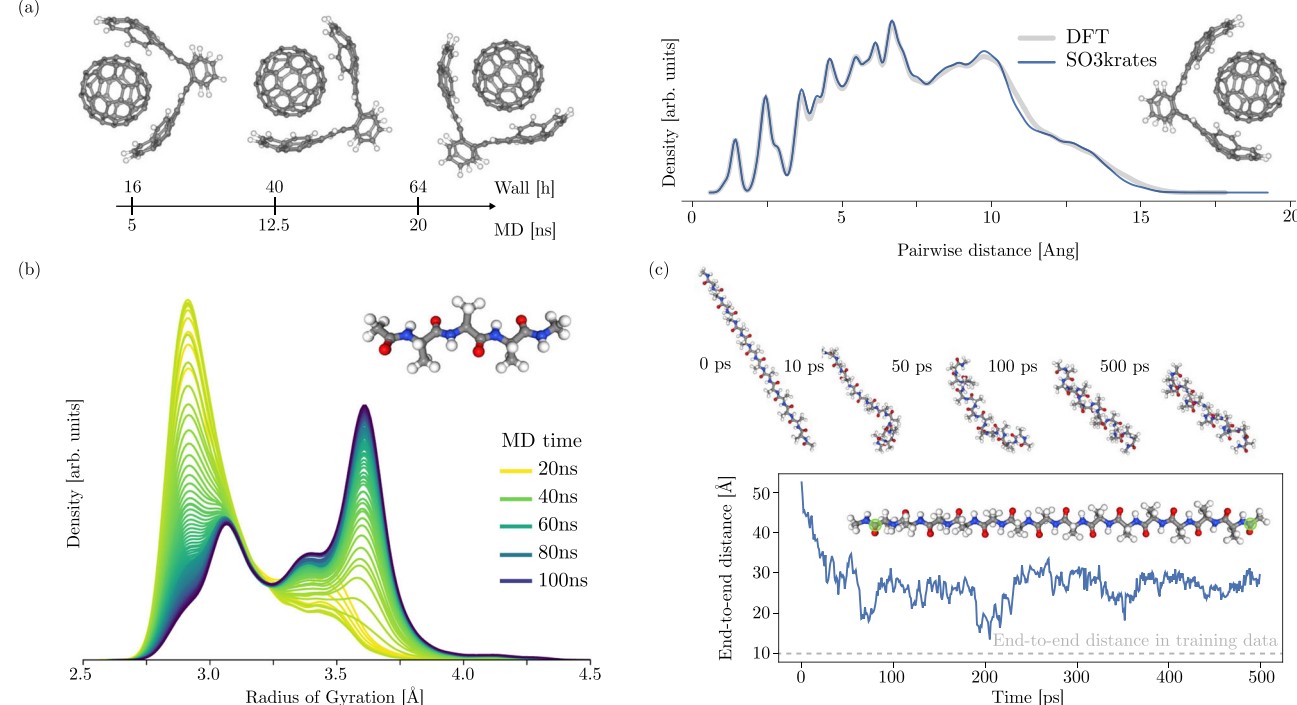

**Fig. 5 | Stable, long-timescale molecular dynamics (MD) simulations and extrapolation to larger bio-molecules. a** Stability and speed of SO3KRATES enable nanosecond-long MD simulations for supra-molecular structures within a few hours. For the buckyball catcher, the ball stays in the catcher over the full simulation time of 20 ns, illustrating that the model successfully picks up on weak, non-covalent bonding. **b** Distribution of the radius of gyration (ROG) for Ac-Ala3-NHMe as a function of MD simulation time in 20 ns steps. The distribution converges after 60–80 ns simulation time, underlining the importance of stable, but at the same time computationally efficient, simulations. In Supplementary Fig. 8 the ROG as a function of simulation time is shown. **c** Dynamics of Ala15 obtained from a SO3KRATES model, trained on only 1k data points of a much smaller peptide (Ac-Ala3-NHMe). Analysis of the end-to-end difference shows rapid folding into helical structure illustrating the generalization capabilities of the learned local representations towards conformational changes on length-scales greatly exceeding the training data (dashed gray line).

until the validation loss converges. Afterwards, we train the equivariant model towards the same validation error, which leads to identical errors on the unseen test set (Fig. 4d, and Supplementary Table 1). Since the equivariant model makes more efficient use of the training data, it requires only ~1/5 of the number of training steps of an invariant model to reach the same validation error (Supplementary Fig. 4d).

After training, we compare the test error distributions since identical mean statistics do not imply a similar distribution. We calculate per atom force errors as $\epsilon_i = ||\vec{F}_i^{\text{pred}} - \vec{F}_i^{\text{ref}}||_2$ and compare the resulting distribution of the invariant and the equivariant model. The so observed distributions are identical in nature and only differ slightly in height and spread without the presence of a clear trend (Supplementary Fig. 5b). In the distributions of the per-structure $\mathcal{S}$ force error $R_i = \frac{1}{|\mathcal{S}|}\sum_{i\in\mathcal{S}}\epsilon_i$, however, one finds a consistently larger spread of the error (Fig. 4d). Thus, the invariant model performs particularly well (and even better than the equivariant model) on certain conformations which comes at the price of worse performance for other conformations, a fact which is invisible to per-atom errors.

The stability coefficients (Eq. (28)) are determined from six 300 ps MD simulations with a time step of 0.5 fs at temperatures $T = 300$ K and $T = 500$ K (Fig. 4d). We find the invariant model to perform best on Ac-Ala3-NHMe, which is the smallest and less flexible structure of the three under investigation where one observes a noticeable decay in stability for the larger temperature. Due to the increase in temperature, configurations that have not been part of the training data are visited more frequently, which requires better extrapolation behavior. When going to flexible structures such as DHA (second row Fig. 4 (d)) the invariant model becomes unable to yield stable MD simulations. To exclude the possibility that the instabilities in the invariant case are due

to the SO3KRATES model itself, we also trained a SCHNET model which yielded MD stabilities comparable to the invariant SO3KRATES model. Thus, directional information has effects on the learned energy manifold that go beyond accuracy and data efficiency.

A subtle case is highlighted by the adenine-thymine complex (AT-AT). The MD simulations show one instability (in a total of six runs) for the equivariant model at 500 K, which illustrates that the stability improvement of an equivariant model should be considered as a reduction of the chance of failure rather than a guarantee for stability. We remark that unexpected behaviors can not be ruled out for any empirical model. We further observed dissociation of substructures (either A, T or AT) from the AT-AT complex during MD simulations (Supplementary Fig. 6). Such a behavior corresponds to the breaking of hydrogen bonds or $\pi$-$\pi$-interactions, which highlights weak interactions as a challenge for MLFFs. Interestingly, for other supra-molecular structures the non-covalent interactions are described correctly (section II F and Fig. 5a). The training data for AT-AT has been sampled from a 20 ps long ab-initio MD trajectory which only covers a small subset of all possible conformations and makes it likely to leave the data manifold. As a consequence, we observe an increase in the rate of dissociation when increasing the simulation temperature, since it effectively extends the space of accessible conformations per unit simulation time.

### Radius of gyration

The radius of gyration (ROG) is an important observable for determining the structural and dynamical behavior of polymers as it allows to gain an estimate for structural compactness of proteins[58] and is experimentally accessible[59]. The timescales of structural changes are often between tenths or even hundreds of nanoseconds[60] which

requires simulation durations at the same order of magnitude to observe a converged distribution of the ROG. Thus, the MLFF must be robust enough to yield stable dynamics for hundreds of nanoseconds while being computationally efficient at the same time.

Here we showcase the potential of the proposed SO3<sub>KRATES</sub> model for such applications by using it to calculate a converged distribution of the ROG for Ac-Ala3-NHMe from 100 ns long MD simulations at 300 K (Fig. 5b). With a time step of 0.5 fs such a simulation requires 200M force evaluations. The SO3<sub>KRATES</sub> model enables us to perform such simulations within 5 days on a single A100 GPU. By analyzing the ROG distribution as a function of simulation time (different colors in Fig. 5b) we find such timescales to be necessary for convergence. We find characteristic peaks in the distribution, which correspond to the folded and unfolded conformation, respectively. Details on the MD simulation can be found in the "Methods" section.

### Generalization to larger peptides

Generalization to larger structures and unknown conformations is an inevitable requirement for scaling MLFFs to realistic simulations in biochemistry. Here, we showcase that by only using 1k training points of a small peptide (42 atoms), SO3<sub>KRATES</sub> can generalize to much larger peptides (151 atoms) without the need of any additional training data. Despite the locality of the model, we observe folding into a helical structure, illustrating the extrapolation capabilities of the learned representations to larger structures.

We use the same model as for the ROG experiment from section II D. As already illustrated, the obtained model is able to perform long and stable dynamics for the structure it has been trained on. To further increase the complexity of the task, we use the model without any modification to investigate the dynamics of Ala15, starting from the extended structure (most left structure in Fig. 5c). Our analysis of the end-to-end distance between the carbonyl carbon of the first residue and the last residue (green spheres in Fig. 5c) reveals that the peptide rapidly folds into the secondary, helical structure (most right structure in Fig. 5c). A comparison to the end-to-end distance in Ac-Ala3-NHMe (dashed horizontal line) reveals the generalization capabilities of SO3<sub>KRATES</sub> towards conformational changes on length-scales that go beyond the ones present in the training data (gray dashed line in Fig. 5c).

### Power spectra

Power spectra of the atom velocities are an important tool to relate MD simulations to real world experimental data. They are calculated as the Fourier transform of the velocity auto-correlation function for systems ranging from small peptides up to host-guest systems and nanostructures. To achieve a correct description for such systems, the model must describe both covalent and non-covalent bonding correctly. For the largest structure with 370 atoms, 5M MD steps with SO3<sub>KRATES</sub> takes 20 h simulation time ( ~15 ms per step).

We train an individual model for each structure in the MD22 data set and compare it to the sGDML model (Table 3). A comparison of training time to other neural network architectures is given in the "Methods" section IV I. To that end, we decided to train the model on two different sets of training data sizes: (A) On structure depended sizes (600 to 8k) as reported in[61], and (B) on structure independent sizes of 1k training points per structure. Since some settings might require accurate predictions when trained on a smaller number of training data points, we chose to include setting (B) into our analysis. The approximation accuracies achievable with SO3<sub>KRATES</sub> compare favorably to the ones that have been observed with the sGDML model[35,61] (Table 3). Even for setting (B) the force errors on the test set are below 1 kcal mol$^{-1}$ Å$^{-1}$. We use the SO3<sub>KRATES</sub> FF to run 1 ns long MD simulations, from which we calculate the power spectra, enabling a comparison to experimental data from IR spectroscopy. Although the frequencies in these systems do not require such simulation lengths,

we chose them to illustrate computational feasibility as well as simulation stability. We start by analyzing two supra-molecular structures in form of a host-guest system and a small nanomaterial. The former play an important role for a wide range of systems in chemistry and biology[26,62], whereas the latter offer promises for the design of materials with so far unprecedented properties[63]. Here, we investigate the applicability of the SO3<sub>KRATES</sub> FF to such structures on the example of the buckyball catcher and the double walled nanotube (Fig. 6a).

For both systems under investigation, one finds notable peaks for C-C vibrations (500 cm$^{-1}$ and 1500 cm$^{-1}$), C-H bending ( ~900 cm$^{-1}$) and for high frequency C-H stretching ( ~3000 cm$^{-1}$). Both systems exhibit covalent and non-covalent interactions[62,64], where e.g., van-der-Waals interactions hold the inner tube within the outer one. Although small in magnitude, we find the MLFF to yield a correct description for both interaction classes, such that the largest degree of freedom for the double walled nanotube corresponds to the rotation of the tubes w. r. t. each other, in line with the findings from[61].

For DHA, we further analyze the evolution of the power spectrum with temperature and find non-trivial shifts in the spectrum hinting towards the capability of the model to learn non-harmonic contributions of the PES (6b). As pointed out in[65], FFs that only rely on (learn) harmonic bond and angle approximations fail to predict changing population or temperature shifts in the middle to high frequency regime. To further showcase the anharmonicity of the spectra obtained with SO3<sub>KRATES</sub>, we first identify the global minimum of DHA (using the minima hopping results from section II G). For the found conformation, we calculate harmonic frequencies as well as the zero point energy (ZPE) from harmonic approximation. The ZPE is 12.979 eV, which corresponds to a temperature of roughly 930 K. We find that our method yields stable dynamics, even though the simulation temperature is almost twice as high as the one used for generating the training data (500 K). The emergence of non-trivial shifts between the spectra from the NVE (blue curve) with the harmonic frequencies (dashed red lines) illustrates the non-trivial anharmonicity that our proposed method is able to model. Similar results are obtained for Ac-Ala3-NHMe (Supplementary Fig. 2).

### Potential energy surface topology

The accurate description of conformational changes remains one of the hardest challenges in molecular biophysics. Every conformation is associated with a local minimum on the PES, and the count of these minima increases exponentially with system size. This limits the applicability of ab-initio methods or computationally expensive MLFFs, since even the sampling of sub-regions of the PES involves the calculation of thousands to millions of equilibrium structures. Here, we explore 10k minima for two small bio-molecules, which requires ~30M FF evaluations per simulation. This analysis would require more than a year with DFT and more than a month with previous equivariant architectures, whereas we are able to perform it in ~2 days.

We employ the minima hopping algorithm[46], which explores the PES based on short MD simulations (escapes) that are followed by structure relaxations. The MD temperature is determined dynamically, based on the history of minima already found. In that way low energy regions are explored and high energy (temperature) barriers can be crossed as soon as no new minima are found. This necessitates a fast MLFF, since each escape and structure relaxation process consists of up to a few thousands of steps. At the same time, the adaptive nature of the MD temperature, can result in temperatures larger than the training temperature (Supplementary Fig. 9a) which requires stability towards out-of-distribution geometries.

We start by exploring the PES of DHA and analyze the minima that are visited during the optimization (Fig. 6c). We find many minima close in energy that are associated with different foldings of the carbon chain due to van-der-Waals interactions. This is in contrast to the minimum energies found for other chainlike molecules such as

## Table 3 | Performance on MD22

| | | Ac-Ala3-NHMe | DHA | Stachyose | AT-AT | AT-AT-CG-CG | Buckyball catcher | Double walled nanotube |
|---|---|---|---|---|---|---|---|---|
| # training points | | 6k | 8k | 8k | 3k | 2k | 600 | 800 |
| sGDML | Energy | 0.39 | 1.29 | 4.00 | 0.72 | 1.42 | 1.17 | 4.00 |
| | Forces | 0.79 | 0.75 | 0.68 | 0.69 | 0.70 | 0.68 | 0.52 |
| SO3krates | Energy | 0.337 | 0.379 | 0.442 | 0.178 | 0.345 | 0.381 | 0.993 |
| | Forces | 0.244 | 0.242 | 0.435 | 0.216 | 0.332 | 0.237 | 0.727 |
| # training points | | 1k | 1k | 1k | 1k | 1k | 1k | 1k |
| SO3krates | Energy | 0.270 | 0.338 | 0.571 | 0.237 | 0.387 | 0.343 | 1.171 |
| | Forces | 0.417 | 0.363 | 0.623 | 0.310 | 0.404 | 0.224 | 0.761 |

We report mean absolute errors (MAEs) for the recently introduced MD22 benchmark and compare it to the sGDML results. Additionally, we report results for a constant number of 1k training points. Units for energy and forces are kcal mol$^{-1}$ and 1 kcal mol$^{-1}$ Å$^{-1}$.

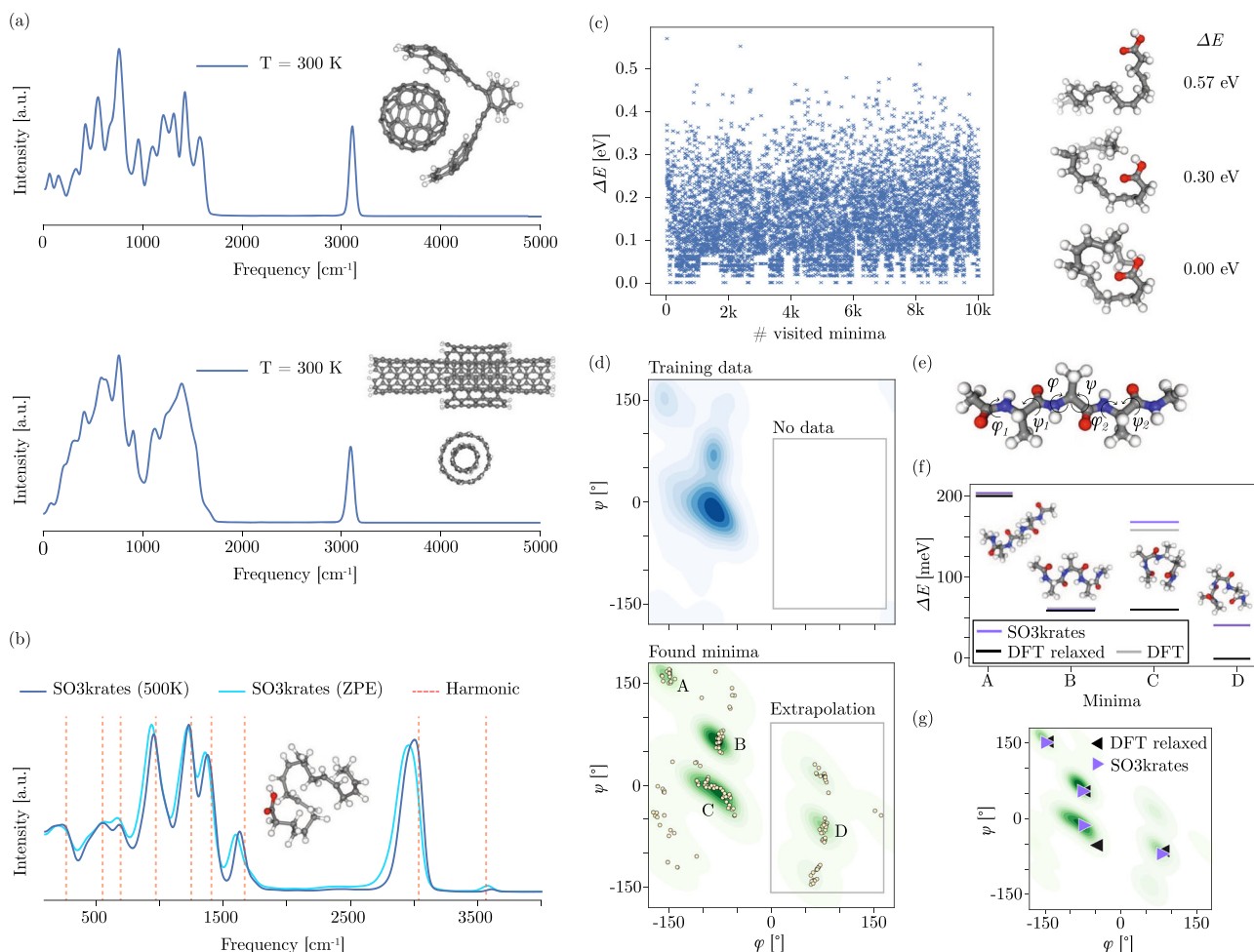

**Fig. 6 | Power spectra and potential energy surface exploration via minima hopping. a** Power spectra for the buckyball catcher (upper panel) and the double walled nanotube (lower panel) computed as the Fourier transform of the velocity auto-correlation function. For the nanotube, the structure is shown from the side and from the front. **b** Power spectrum for DHA from an NVE at the zero point energy (ZPE) (light blue) and 500 K (dark blue), as well as the frequencies from harmonic approximation (red dashed lines). **c** Results of a minima search for DHA. We ran the simulation until 10k minima had been visited, which corresponds to 20 M molecular dynamics steps for the escape trials and to ~10 M PES evaluations for the structure relaxations, afterwards. Next to it minima with the largest energy (top), the lowest energy (bottom) and an example minimum with an intermediate energy value (middle) are depicted. **d** Ramachandran density plots for the training

conformations (upper, blue) and of the visited minima during minima hopping (lower, green) for two of the six backbone angles in Ac-Ala3-NHMe. Yellow dots correspond to the actually visited minima. Parts of the visited minima have not been in the training data, hinting towards the capability of the model to find minima beyond the training data. **e** Ac-Ala3-NHMe structure with backbone angles as inset. **f** Relative energies for four minima, which have been selected from the regions in $\psi - \phi$ space visited most frequently during minima hopping (A–D in (**d**)). SO3krates energies are compared to a DFT single point calculation and to the conformation obtained from a full DFT relaxation starting from the minima obtained from SO3krates. **g** Location in the Ramachandran plot of the minima obtained with SO3krates and the relaxed DFT minima.

Ac-Ala3-NHMe, where less local minima are found per energy unit (Supplementary Fig. 11a). The largest observed energy difference corresponds to 0.57 eV, where the minima with the largest potential energy (top) and the lowest potential energy (bottom) as well as an example structure from the intermediate energy regime (middle) are shown in Fig. 6c). We find the observed geometries to be in line with the expectation that higher energy configurations promote an unfolding of the carbon chain.

Funnels are sets of local minima separated from other sets of local minima by large energy barriers. The detection of folding funnels plays an important role in protein folding and finding native states, which determine the biological functioning and properties of proteins. The combinatorial explosion of the number of minima configurations makes funnel detection unfeasible with ab initio methods or computationally expensive MLFFs. We use the visited minima and the transition state energies that are estimated from the MD between successive minima to create a so-called *disconnectivity graph*[66]. It allows detect multiple funnels in the PES of DHA, which are separated by energy barriers up to 3 eV (Supplementary Fig. 12).

Ac-Ala-NHMe is a popular example system for bio-molecular simulations, as its conformational changes are primarily determined by Ramachandran dihedral angles. These dihedral angles also play a crucial role in representing important degrees of freedom in significantly larger peptides or proteins[67]. Here, we go beyond this simple example and use the minima hopping algorithm to explore 10k minima of Ac-Ala3-NHMe and visualize their locations in a Ramachandran plot (green in Fig. 6d) for two selected backbone angles $\phi$ and $\psi$ (Fig. 6e). By investigating high-density minima regions and comparing them to the training data (blue in 6d), we can show that SO3KRATES finds minima in PES regions, which highlights the capability of the model to extrapolate beyond known conformations. Extrapolation to unknown parts of the PES is inevitable for the application of MLFFs in bio-molecular simulations, since the computational cost of DFT only allows to sample sub-regions of the PES for increasingly large structures.

To confirm the physical validity of the found minima, we select one equilibrium geometry from each of the four highest density regions in the Ramachandran plots (A–D in Fig. 6d). A comparison of the corresponding energies predicted by SO3KRATES with DFT single point calculations (Fig. 6f) shows excellent agreement with a mean deviation of 3.45 meV for this set of four points. Remarkably, the minimum in the unsampled region of the PES (gray box in Fig. 6d) only deviates by a mere 0.7 meV in energy. We further compare the SO3KRATES relaxed structure to structures obtained from a DFT relaxation, initiated from the same starting points. For minima A and B, we again find excellent agreement with an energy error of 2.38 meV and 3.57 meV, respectively. The extrapolated minima D shows a slightly increased deviation (41.84 meV), which aligns with our expectation that the model performs optimally within the training data regime. Further, minima A, B, and D show good agreement with the backbone angles obtained from DFT relaxations (Fig. 6g).

For minimum C, we find the largest energy deviation w. r. t. both, DFT single point calculation and DFT relaxation. When comparing the relaxed structures, we observe a rotation of 180° for one methyl group, the addition of a hydrogen bond and a stronger steric strain in the SO3KRATES prediction. These deviations coincide with a relatively large distance in the $\phi$-$\psi$ plane (Fig. 6g). To investigate the extend of minimum 3, we have generated random perturbations of the equilibrium geometry from which additional relaxation runs have been initiated. All optimizations returned into the original minimum (Supplementary Fig. 11b), confirming that it is not an artifact due to a non-smooth or noisy PES representation.

## Discussion

Long-timescale MD simulations are essential to reveal converged dynamic and thermodynamic observables of molecular systems[60,68–71].

Despite achieving low test errors, many state-of-the-art MLFFs exhibit unpredictable behavior caused by the accumulation of unphysical contributions to the output, making it extremely difficult or even impossible to reach extended timescales[32]. This prevents the extraction of physically faithful observables at scale. Ongoing research aims at improving stability by incorporating physically meaningful inductive biases via various kinds of symmetry constraints[8,11,17,36,37,72,73], but the large computational cost of current solutions mitigates many practical advantages.

We overcome the challenging trade-off between stability and computational cost by combining two concepts—a Euclidean self-attention mechanism and the EV as efficient representation for molecular geometry—within the equivariant transformer architecture SO3KRATES. The exceptional performance of our approach is due to the decoupling of invariant and equivariant information, which enables a substantial reduction in computational complexity compared to other equivariant models.

Our architecture strategically emphasizes the importance of the more significant invariant features over equivariant ones, resulting in a more efficient allocation of computational resources. While equivariant features carry important directional information, the core of ML inference lies in the invariant features. Only invariant features can be subjected to powerful non-linear transformations within the architecture, while equivariant features essentially have to be passed-through to the output in order to be preserved. In our implementation, the computationally cheap invariant parts ($l = 0$) of the model are allowed to use significantly more parameters than the costly equivariant ones ($l > 0$). Despite this heavy parameter reduction of the equivariant components, desirable properties associated with equivariant models, such as high data efficiency, reliable MD stability, and temperature extrapolation, could still be preserved.

In the context of MD simulations, we found that the equivariant network (SO3KRATES with $l_{\max} > 0$) gives smaller force error distributions than its invariant counterpart (SO3KRATES with $l_{\max} = 0$). This effect, however, is only visible when the force error is investigated on a per-structure and not on the per-atom level. This observation indicates that the invariant network over-fits to certain structures. We also found the equivariant model to remain stable across a large range of temperatures, whereas the stability of the invariant model quickly decreases with increasing temperature. Since higher temperatures increase the probability of out-of-distribution geometries, this may hint towards a better extrapolation behavior of the equivariant model.

Applying the SO3KRATES architecture to different structures from the MD22 benchmark, including peptides (Ac-Ala3-NHMe, DHA) and supra-molecular structures (AT-AT, buckyball catcher, double walled nanotube), yields stable molecular dynamics (MD) simulations with remarkable timescales on the order of tens of nanoseconds per day. This enables the computation of experimentally relevant observables, including power spectra and converged distributions for the radius of gyration for small peptides. We have also shown, that SO3KRATES reliably reveals conformational changes in small bio-molecules on the example of DHA and Ac-Ala3-NHMe. To that end, SO3KRATES is able to predict physically valid minima conformations that have not been part of the training data. The representative nature of Ac-Ala3-NHMe holds the potential that a similar behavior can be obtained for much larger peptides and proteins. The limited availability of ab-initio data for structures at this scale, makes extrapolation to unknown parts of the PES a crucial ingredient on the way to large scale bio-molecular modeling.

In a recent line of work, methods have been proposed that aim to reduce the computational complexity of SO(3) convolutions[74,75]. They serve as a drop-in replacement for full SO(3) convolutions whereas the method presented here allows to fully avoid expensive SO(3) convolutions within the message passing paradigm. This result as well as ours demonstrates that the optimization of equivariant interaction is

an active research field that has not yet fully matured, potentially offering further paths for improvement.

While our development makes stable extended simulation time-scales accessible using modern MLFF modeling paradigms, future work remains to be done in order to bring the applicability of MLFFs even closer to that of conventional classical FFs. Various encouraging avenues in that direction are currently emerging: In the current design, the EV are only defined in terms of two-body interactions. Recent results suggest that accuracy can be further improved by incorporating atomic cluster expansions into the MP step[52,76–78]. At the same time, this may help reduce the number of MP steps which in turn decreases the computational complexity of the model.

Another yet open discussion is the appropriate treatment of global effects. Promising steps have been taken by using low-rank approximations[24,79], trainable Ewald summation[80] or by learning long-range corrections in a physically inspired way[17,24,81–87]. Approaches of the latter type are of particular importance when extrapolation to larger systems is required. Although equivariant models can improve the extrapolation capabilities for local interactions, this does not hold for interactions that go beyond the length scales present in the training data or beyond the effective cutoff of the model. Since the aforementioned methods rely on local properties such as partial charges[17,24,83,84], electronegativities[85] or Hirshfeld volumes[81,82] they can be seamlessly integrated into our method by learning a corresponding local descriptor in the invariant feature branch of the SO3KRATES architecture.

Future work will, therefore, focus on the incorporation of many-body expansions, global effects, and long-range interactions into the EV formalism and aim to further increase computational efficiency to ultimately bridge MD timescales at high accuracy.

## Methods
### Features and Euclidean variables (EV)
Per-atom feature representations are initialized based on the atomic number $z_i$ using an embedding function

$$f_i = f_{\text{emb}}(z_i),$$ (10)

which maps the atomic number into the $F$ dimensional feature space $f_{\text{emb}} : \mathbb{N}_+ \mapsto \mathbb{R}^F$.

For a given degree $l$ and order $m$, the EV are defined as

$$x_{ilm} = \frac{1}{\langle \mathcal{N} \rangle} \sum_{j \in \mathcal{N}(i)} \phi_{r_{\text{cut}}}(r_{ij}) \cdot Y_m^l(\hat{r}_{ij}),$$ (11)

where the output of $Y_m^l(\hat{r}_{ij})$ is modulated with a distance dependent cutoff function which ensures a smooth PES when atoms leave or enter the cutoff sphere. Alternatively, the EV can be initialized with all zeros, such that they are "initialized" in the first attention update (16). The aggregation is re-scaled by the average number of neighbors over the whole training data set $\langle \mathcal{N} \rangle$, which helps stabilize network training. By collecting all degrees and orders up to $l_{\text{max}}$ within one vector

$$\boldsymbol{x}_i = \left[ x_{i00}, x_{i1-1}, \ldots, x_{il_{\text{max}} l_{\text{max}}} \right],$$ (12)

one obtains an equivariant per-atom representation of dimension $(l_{\text{max}} + 1)^2$ which transforms according to the corresponding Wigner-D matrices.

### SO(3) Convolution Invariants
The convolution output for degree $L$ and order $M$ on the difference vector $\boldsymbol{x}_{ij} = \boldsymbol{x}_j - \boldsymbol{x}_i$ can be written as

$$x_{ij}^{LM} = \sum_{l_1 l_2 m_1 m_2} C_{l_1 l_2 L}^{m_1 m_2 M} x_{ij}^{l_1 m_1} x_{ij}^{l_2 m_2}$$ (13)

where $C_{l_1 l_2 L}^{m_1 m_2 M}$ are the Clebsch-Gordan coefficients. Considering the projection on the zeroth degree $L = M = 0$

$$x_{ij}^{00} = \sum_{l_1} \underbrace{\sum_{m_1} C_{l_1 l_1 0}^{m_1 - m_1 0} x_{ij}^{l_1 m_1} x_{ij}^{l_1 - m_1}}_{\equiv \bigoplus_{l=0}^{l_{\text{max}}} \boldsymbol{x}_{ij,l \to 0}},$$ (14)

one can make use of the fact that $C_{l_1 l_2 L}^{m_1 m_2 M}$ is valid for $|l_1 - l_2| \le L \le l_1 + l_2$ and $M = m_1 + m_2$, which corresponds to having nonzero values along the diagonal only (Fig. 2). Thus, evaluating $\bigoplus_{l=0}^{l_{\text{max}}} \boldsymbol{x}_{ij,l \to 0}$ requires to take per-degree traces of length $(2l + 1)$ and can be computed efficiently in $\mathcal{O}(l_{\text{max}}^2)$.

### Euclidean transformer block (ECTBLOCK) and Euclidean self-attention
Given input features, EV and pairwise distance vectors the Euclidean attention block returns *attended* features and EV as

$$f_i^{\text{ATT}} = f_i + \sum_{j \in \mathcal{N}(i)} \phi_{r_{\text{cut}}}(r_{ij}) \cdot \alpha_{ij} \cdot f_j,$$ (15)

and

$$x_{ilm}^{\text{ATT}} = x_{ilm} + \sum_{j \in \mathcal{N}(i)} \phi_{r_{\text{cut}}}(r_{ij}) \cdot \alpha_{ijl} \cdot Y_l^m(\hat{r}_{ij}),$$ (16)

with a cosine cutoff function

$$\phi_{r_{\text{cut}}}(r_{ij}) = \frac{1}{2} \left( \cos\left( \frac{\pi r_{ij}}{r_{\text{cut}}} \right) + 1 \right),$$ (17)

which guarantees that pairwise interactions (attention coefficients) smoothly decay to zero when atoms enter or leave the cutoff radius $r_{\text{cut}}$. Eqs. (15) and Eq. (16) from above involve attention coefficients which are constructed from an equivariant attention operation (next paragraphs).

Attention coefficients are calculated as

$$\alpha_{ij} = \alpha \left( f_i, f_j, g_{1,\ldots K}(r_{ij}), \bigoplus_{l=0}^{l_{\text{max}}} \boldsymbol{x}_{ij,l \to 0} \right),$$ (18)

where $\boldsymbol{x}_{ij} \equiv \boldsymbol{x}_j - \boldsymbol{x}_i \in \mathbb{R}^{(l_{\text{max}} + 1)^2}$ is a relative, higher order geometric shift between neighborhoods. The function $\bigoplus_{l=0}^{l_{\text{max}}} \boldsymbol{x}_{ij,l \to 0}$ contracts each degree in $\boldsymbol{x}_{ij}$ to the zeroth degree which results in $l_{\text{max}} + 1$ invariant scalars (Eq. (14)). The function $g$ expands interatomic distances in $K$ radial basis functions (RBFs)

$$g_k(r_{ij}) = \exp\left( -\gamma (\exp(-r_{ij}) - \mu_k)^2 \right),$$ (19)

where $\mu_k$ is the center of the $k$-th basis function and $\gamma$ is a function of $K$ and $r_{\text{cut}}$[17].

Based on the output of the contraction function and the RBFs we construct an $F$-dimensional filter vector as

$$w = \text{MLP}_{[F/4, F]}(u) + \text{MLP}_{[F,F]}(g),$$ (20)

where $\text{MLP}_{[F_1, \ldots F_L]}$ denotes a multi layer perceptron network with $L$ layers, layer dimension $F_i$ and SILU non-linearity. The first MLP acting on $u$ has a reduced dimension in the first hidden layer (since the dimension of $u$ itself is only $l_{\text{max}} + 1$).

Attention coefficients are then calculated using dot-product attention as

$$\alpha_{ij} = \frac{1}{\sqrt{F}} q_i^T (w_{ij} \odot k_j),$$ (21)

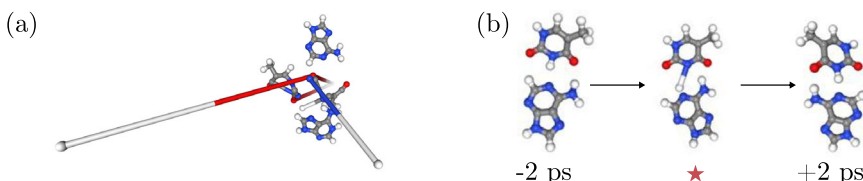

**Fig. 7 | Potential instabilities that can occur in a molecular dynamics (MD) simulation using machine learning force fields. a** Illustration of an "explosion" during an MD simulation. **b** Illustration of a temporarily limited instability (here the breaking of a covalent bond). The point of time of the instability is marked via a red cross.

where $\odot$ denotes the entry-wise product and $q_i = Qf_j$ and $k_j = Kf_j$ with $K \in \mathbb{R}^{F \times F}$ and $Q \in \mathbb{R}^{F \times F}$ are trainable key and query matrices. The attention update of the features (Eq. (15)) is performed for $h$ heads in parallel. The features $f_i$ of dimension $F$ are split into $h$ feature heads $f_i^h$ of dimension $(h, F/h)$. From each feature head, one attention coefficient $\alpha_{ij}^h$ is calculated following Eq. (18) where $q_i$, $k_j$ and $w_{ij}$ are all of dimension $F/h$. For each head, the attended features are then calculated from Eq. (15) with $f_i$ replaced by the corresponding head $f_i^h$. Afterwards, the heads are stacked to form again a feature vector of dimension $f_i$. Multi-head attention allows the model to focus on different sub-spaces in the feature representation, e.g., information about distances, angles or atomic types[44].

**Interaction Block**
An interaction block (IBLOCK) aims to interchange per-atom information between the invariant and the geometric variables. Refinements for invariant features and equivariant EV are calculated as

$$\mathrm{d}f_i, \mathrm{d}\boldsymbol{x}_i = \mathrm{IBLOCK}\left(f_i^{\mathrm{ATT}}, \oplus_{l=0}^{l_{\max}} \boldsymbol{x}_{i,l\to 0}^{\mathrm{att}}\right), \qquad (22)$$

More specifically, the refinements are calculated as

$$\mathrm{d}f_i = a \qquad (23)$$

and

$$\mathrm{d}x_{ilm} = b_l x_{ilm}^{\mathrm{ATT}}, \qquad (24)$$

where $a \in \mathbb{R}^F$ and one $b_l \in \mathbb{R}$ for each degree $l$. They are calculated from a singled layered MLP as

$$a, b = \mathrm{MLP}_{[f + l_{\max} + 1]}\left(f_i^{\mathrm{ATT}}, \oplus_{l=0}^{l_{\max}} \boldsymbol{x}_{i,l\to 0}^{\mathrm{att}}\right) \qquad (25)$$

such that $a$ and $b = [b_0, \ldots, b_{l_{\max}}]$ contain mixed information about both $f_i$ and $\boldsymbol{x}_i$. Updates are then calculated as

$$f_i^{[t+1]} = f_i^{\mathrm{ATT}} + \mathrm{d}f_i, \qquad (26)$$

$$\boldsymbol{x}_i^{[t+1]} = \boldsymbol{x}_i^{\mathrm{ATT}} + \mathrm{d}\boldsymbol{x}_i, \qquad (27)$$

which builds the relation to the initially stated update equations of the ECTBLOCK and concludes the architecture description.

**MD stability**
We define an MD simulation to be stable when (A) there is no non-physical dissociation of the system, and (B) each bond length follows a reasonable distribution over time. We refer to failure mode (A) when there is a divergence in the predicted velocities resulting in instantaneous dissociation of all or some atoms in the molecule which corresponds to non-physical behavior (Fig. 7a). In contrast, a physically meaningful dissociation happens on reasonable time and kinetic energy scales. A decomposition of (parts of) the molecule can be detected by a strong peak in MD temperature, which is usually a few

orders of magnitude larger than the target temperature. We assume a bond length to be distributed reasonably, when it does not differ by more than 50 % from the equilibrium bond distance at any point of the simulation. Criteria (A) has e.g., been used in[32] to determine the MD stability of different MLFFs. However, in certain cases analyzing MD temperature can be an insufficient condition to detect unstable behavior, e.g., when single bonds dissociate slowly over time or take on non-physical values over a temporarily limited interval (Fig. 7b). Such a behavior, however, is easily identified using criteria (B).

A stability coefficient $c_s \in [0, 1]$ is then calculated as

$$c_s = \frac{n_s}{n_{\mathrm{tot}}}, \qquad (28)$$

where $n_s$ is the number of MD steps until an instability occurs and $n_{\mathrm{tot}}$ is the maximal number of MD steps. When no instability is observed in the simulations we set $n_s = n_{\mathrm{tot}}$.

**MD Simulations**
For the MD simulation of $Li_3PO_4$ we chose the first conformation in the quenched state as initial starting point. We then run the simulation for 50 ps with a time step of 2 fs using a Nose-Hoover thermostat at 600 K. For MD simulations with molecules from the MD22 data set we first chose a structure which has not been part of the training data. It is then relaxed using the LBFGS optimizer until the maximal force norm per atom is smaller than $10^{-4}$ eVÅ$^{-1}$. The relaxed structure serves as starting point for the MD simulation. For the comparison of invariant and equivariant model, we run three MD simulations per molecule from three different initial conformations with a time step of 0.5 fs and a total time of 300 ps using the Velocity Verlet algorithm without thermostat. For the radius of gyration and Ala15 experiments we performed simulations with 0.5 fs time step at a temperature $T = 300$ K with the system coupled to a heat bath via a Langevin thermostat[88] with a friction coefficient of $\gamma = 10^{-3}$. For the calculation of the spectra, we ran MD simulations with a time step of 0.2 fs following[61] and a total time of 1 ns for the buckyball catcher and the nanotube and 200 ps for the MD of DHA at the zero point energy. Temperatures vary between molecules and are reported in the main body of the text. Again only the Velocity Verlet without thermostat is used. We show in the Supplementary information, that the performed simulations are energy conserving and reach temperature equilibrium. When using the Velocity Verlet algorithm, initial velocities are drawn from a Maxwell Boltzmann distribution with a temperature twice as large as the MD target temperature. For the MD stability experiments on the MD17 molecules, we follow[32] and run simulations with a Nose-Hoover thermostat at 500 K and a time step of 0.5 fs for 300 ps.

**Minima hopping algorithm**
For the minima hopping experiments we use the models that have been trained on the MD22 data set with 1k training samples. Each escape run corresponds to a 1 ps MD simulation with a time step of 0.5 fs using the Velocity Verlet algorithm. The following structure relaxation is performed using the LBFGS optimizer until the maximal norm per atomic force vector is smaller than $10^{-4}$ eVÅ$^{-1}$, which took

around 1k optimizer steps on average. The initial velocities are drawn from the Maxwell-Boltzmann distribution at temperature $T_0$, which are re-scaled afterwards such that the systems temperature matches $T_0$ exactly. Since the structure is in a (local) minima at initialization, the equipartition principle will result in an MD which has temperature $T_0/2$ on average. After the MD escape run the newly proposed minima is compared to the current minima as well as to all the minima that have been visited before (history). Minima are compared based on their RMSD. To remove translations, we compare the coordinates relative to the center of mass. Also, since structures might differ by a global rotation only, we minimize the RMSD over SO(3), following the algorithm described in section 7.1.9 *Rotations* (p. 246–250) in[89]. If RMSD≤$10^{-1}$ between two minima, they are considered to be identical. If the newly proposed minima is not the current minima (i.e., it is either completely new or in the history), the new minima is accepted if the energy difference is below a certain threshold $E_{\text{diff}}$.

For the initial temperature we chose $T_0 = 1000$ K and for $E_{\text{diff}} = 2$ eV. Both quantities are dynamically adjusted during runtime, where we stick to the default parameters[46]. The development of $T_0$ along the number of performed escape runs shows initial temperatures ranging from ~300 K up to ~1300 K (Supplementary Fig. 9b). To estimate the transition states for the connectivity graph, the largest potential energy observed between two connected minima is taken for its energy (Supplementary Fig. 9b).

### Network and training

All SO3KRATES models use a feature dimension of $F = 132$, $h = 4$ heads in the invariant MP update and $r_{\text{cut}} = 5$ Å. The number of MP updates and the degrees in the EV vary between experiments. For the comparison of invariant and equivariant model we use degrees $l = \{0\}$ and $l = \{0, 1, 2, 3\}$, $T = 3$ and EV initialization following Eq. (11). The invariant degree is explicitly included, in order to exclude the possibility that stability issues might come from the inclusion of degree $l = 0$. The number of network parameters of the invariant model is 386k and of the equivariant model is 311k, such that the better stability is not be related to a larger parameter capacity but truly to the degree of geometric information. Due to the use of as many heads as degrees in the MP update for the EV, increasing the number of degrees results in a slightly smaller parameter number for the equivariant model. Per molecule 10,500 conformations are drawn of which 500 are used for validation. For the invariant and equivariant model, two models are trained on training data sets which are drawn with different random seeds. The model for $Li_4PO_3$ uses $T = 2$, $l = \{1, 2, 3\}$ and initializes the EV to all zeros. For training, 11k samples are drawn randomly from the full data set of which 1k are used for validation, following[53].

All other models use degrees $l = \{1, 2, 3\}$ in the EV, $T = 3$ and initialize the EV according to Eq. (11). For the MD17 stability experiments, 10,000 conformations are randomly selected of which 9500 are used for training and 500 for validation. For the MD22 benchmark a varying number of training samples plus 500 validation samples or 1000 training samples plus 500 validation samples are drawn randomly. The models trained on 1000 samples are used for the calculation of the spectra, for the minima hopping experiments and for the radius of gyration analysis.

All models are trained on a combined loss of energy and forces

$$\mathcal{L} = (1 - \beta) \cdot (E - \tilde{E})^2 + \frac{\beta}{3N} \sum_{k=1}^{n} \sum_{i \in (x,y,z)} (F_k^i - \tilde{F}_k^i)^2, \qquad (29)$$

where $\tilde{E}$ and $\tilde{F}$ are the ground truth and $E$ and $F$ are the predictions of the model. We use the ADAM[90] optimizer with an initial learning rate (LR) of $\eta = 10^{-3}$ and a trade-off parameter of $\beta = 0.99$. The LR is decreased by a factor of 0.7 every 100k training steps using exponential LR decay. Training is stopped after 1M steps. The batch sizes $B_s$ for training depends on the number of training points $n_{\text{train}}$,

where we use $B_s = 1$ if $n_{\text{train}} \leq 1000$ and $B_s = 10$ if $n_{\text{train}} \geq 1000$. All presented models can be trained in less than 12 h on a single NVIDIA A100 GPU.

### Training step time

The total training time can strongly vary depending on selection of hyperparameters such as batch size or the number of epochs until convergence of the loss, which itself depends on the number of data points, the learning rate and the optimizer that is used. To make a comparison as fair as possible it thus makes sense to report the time per gradient update when comparing MLFF models.

In ref. 91, the training time consumption has been compared for different state-of-the-art models, where e.g., PAINN[37] takes ~10 ms, VISNET[91] takes ~50 ms and NEQUIP[36] or ALLEGRO[53] take ~200 ms for Chignolin (166 atoms). When training on Chignolin we achieve a training time consumption of ~8 ms measured on an A100 GPU. The training times reported in[91] are measured on a V100 GPU, which is a factor 1.6–3.4 slower than an A100, depending on the data modality and architecture. Thus, our runtime comparison has an uncertainty of the same factor, but can illustrate an order of magnitude faster training step speed compared to models such as NEQUIP or ALLEGRO.

The gradient step time depends on the inference speed of the model as well as on the number of parameters in the model. As already shown SO3KRATES outperforms other equivariant models that are competitive in stability and accuracy in the terms of inference speed. Further, SO3KRATES has only 311 k parameters which is lightweight compared to other equivariant models as ALLEGRO or NEQUIP. Thus, our measured runtimes for training are also in line with our expectation from theoretical considerations.

### Reporting summary

Further information on research design is available in the Nature Portfolio Reporting Summary linked to this article.

## Code availability

The code for SO3KRATES is available at https://doi.org/10.5281/zenodo.11473653[92], which contains interfaces for model training and running MD simulations on GPU.

## Data availability

MD17 data for stability experiments and MD22 data are freely available from http://sgdml.org/#datasets. The $Li_3PO_4$ data can be downloaded from https://archive.materialscloud.org/record/2022.128 and the Chignolin data can be found at https://github.com/microsoft/AI2BMD/tree/ViSNet/chignolin_data. Source data are provided in this paper.

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

## Acknowledgements

JTF, KRM, and SC acknowledge support by the Federal Ministry of Education and Research (BMBF) for BIFOLD (01IS18037A). KRM was partly supported by the Institute of Information & Communications Technology Planning & Evaluation (IITP) grants funded by the Korea government(MSIT) (No. 2019-0-00079, Artificial Intelligence Graduate School Program, Korea University and No. 2022-0-00984, Development of Artificial Intelligence Technology for Personalized Plug-and-Play Explanation and Verification of Explanation), and was partly supported by the German Ministry for Education and Research (BMBF) under Grants 01IS14013A-E, AIMM, 01GQ1115, 01GQ0850, 01IS18025A and 01IS18037A; the German Research Foundation (DFG). The authors would like to thank Niklas Schmitz and Mihail Bogojeski for the helpful discussion. Correspondence to KRM and SC.

## Author contributions

J.T.F. conceptualized the SO3KRATES model. J.T.F. and S.C. prepared the first version of the manuscript. J.T.F. developed the software and performed the numerical experiments. O.T.U. calculated the (relaxed) DFT energies for Ac-Ala3-NHMe. S.C., O.T.U. and K.R.M. supervised and guided the project from conception to design of experiments, theory, as well as analysis of data. All authors contributed to the manuscript.

## Funding

## Competing interests

The authors declare no competing interests.
