## [Peer Review File · Nature Communications]

REVIEWER COMMENTS

Reviewer #1 (Remarks to the Author):

In this paper by Frank et al the authors present the new architecture "SO3krates" to perform better machine learning construction of potential energy surfaces. In particular they aim at demonstrating that they can improve over known machine learned force fields by performing stable molecular dynamics (MD) simulations for a longer time and by investigating larger size systems.

The paper is well written and addresses an important topic in theoretical physics and chemistry. I think that most of data reported are quite impressive and, therefore, the paper is certainly of potential interest for the readership of Nature Communications.

However there is one aspect I think the authors should better clarify and demonstrate. My concern is about what they call the "velocity auto-correlation functions". On this point I think a revision is necessary for the following reasons:

1) What the authors call "velocity auto-correlation function" is actually the Fourier transform of the velocity auto-correlation function. This should be corrected also in the caption of Figure 6.

2) The authors perform molecular dynamics simulations for a total time of 1ns. Looking at the frequencies of vibrations there is no need for such a long time, unless this is the time need for the thermalization step. Is that the reason of such a long evolution?

3) Most importantly the investigated temperatures $T=100\text{K}$, 300K , 500K are likely not high enough to have an energy content comparable to the zero-point energy of the investigated systems. The authors could check this out and calculate the harmonic estimates of the frequencies of vibration (I suggest to do this especially for the CH stretch region around 3000 cm^{-1}). They will probably find harmonic values very close to their dynamical estimate. This is because not enough energy is given to the system. What the authors should do instead is to perform NVE simulations at an energy equal to the zero-point energy of the system. Can they perform such a simulation or does the structure break down because they are investigating regions far away from the training region? If they can perform the simulation, what are the frequency of vibrations? They should find some degree of anharmonicity. If so, then they have strongly demonstrated that their new architecture is really able to reach a whole new level of accuracy. If not, it means their PES is basically harmonic and this should be carefully considered by the authors (there is no need of a very refined approach to have a harmonic PES).

My overall assessment of the paper is certainly positive, but I guess that the authors should properly address the 3 points above before the paper is suitable for acceptance in Nature Communications.

Reviewer #2 (Remarks to the Author):

The paper "From Peptides to Nanostructures: A Euclidean Transformer for Fast and Stable Machine Learned Force Fields" discusses the development of SO3krates, a transformer architecture that combines the benefit of equivariant representations that usually lead to higher stability in ML-driven molecular dynamics, with the computational efficiency of less accurate representations. The way this is achieved is by replacing the $SO(3)$ convolutions and avoiding expensive tensor products by using a filter on the relative orientation of atomic neighborhoods. The projection applied reduces the representation in a way that only most relevant invariant components of the equivariant basis functions are included that correspond to partial traces of the product-tensor. The forces are obtained as derivatives of invariant energy models. Self-attention layers are used to decouple invariant and equivariant features within the model, which decreases computational

complexity compared to other equivariant models. A speed up of about 25 could be achieved in molecular dynamics simulations of small peptides to nanostructures compared to state-of-the-art equivariant models like Nequip, Painn or Allegro without sacrificing accuracy. Finally, the model can be used to detect minimum energy conformations using minima hopping, showcasing the potential for extrapolation, which is especially relevant when scaling up to even larger system. I find the manuscript well written, easy to follow and a major contribution to the field. I only have very little comments specified below.

Question:

- 1) It is not fully clear to me whether vectorial and tensorial properties can be directly predicted with SO(3)krates. If the authors could specify this in the text, it would make it more clear.
- 2) Fig. 4c, it would be a nice comparison to see how the RDF looks for invariant models. This would give a better estimate of how close the RDFs really are when comparing ab initio and SO(3)krates.
- 3) MD stability: Do the authors mean that they define an MD to be stable when there is a controlled dissociation? What is meant by uncontrolled dissociation?
- 4) Can the authors comment on the computational costs of training SO(3)krates compared to other equivariant models? I think that this information would be very helpful in addition to the speed up in prediction time.
- 5) The authors mention the potential of the method for extrapolation to larger systems. However, in such systems, the long-range interactions are often not properly captured within the ML representations such that they need to be encountered in non-equivariant models in a "physically-inspired" way, such as it is done in SpookyNet, PhysNet, or Behlers 3rd and 4th generation of NNs or externally via charge predictions (Hirshfeld volumes) as done, e.g., in DOI:10.1039/D2DD00016D or DOI:10.1103/PhysRevB.104.054106. The latter two papers present methods that are also used for materials and hybrid systems including also studies on minima searches using basin hopping algorithms. Can the authors comment on the usage of their method for materials? Do they expect that some additional physics in the model is required or do they expect that their representation captures long-range effects properly?

Typo: page 4, right paragraph, "...by and attention function..."

Reviewer #4 (Remarks to the Author):

This work proposes an equivariant machine learning force field (MLFF) model to address the computational speed and accuracy tradeoff. Results include applications to small molecule dynamics and conformation sampling. The main idea is to sparsify the equivariant tensor product operation by avoiding the full product operation and operating only on its invariant output, there by reducing the scaling with the rotation degree of spherical harmonics.

Overall, the work is interesting, but its impact is limited for several reasons. Proposed architectural modifications to the equivariant neural network architecture are incremental improvements, resembling previous works even if not identically implemented. Moreover, significant advances in "unprecedented" speed and accuracy are claimed but are not systematically demonstrated against state-of-the-art models. Furthermore, only small molecules are considered, and no applications to large more relevant systems are demonstrated, which is the main goal of MLFFs. I do not find this paper appropriate for Nature Communications, but could be a good contribution to a more specialized journal after major revisions.

This is a rapidly moving competitive field, and it is important to avoid claims about model speed, stability and accuracy, unless optimal frontiers for each model architecture are considered and compared fairly. Each model has hyperparameters that affect such tradeoffs in different ways. Therefore, a claim of a significantly better model must be substantiated in a systematic quantitative manner.

Timing tests, especially on small systems, are known to be highly variable and dependent on the ability of NN frameworks to achieve GPU utilization. In addition, the integration of the models into

MD codes can also. This paper does not specify which framework was used, which JIT optimizations were used, how the models were implemented with MD codes, how the timings were measured. The overheads of MD codes may or may not be neglected. Without these details and without systematic sweeps through model hyperparameters to establish accuracy-speed pareto frontiers. At this time in the MLFF field, it is unacceptable to publish a paper with claims of model superiority without exploring these considerations.

The findings that disabling equivariance leads to less stable models, due to worse extrapolation, wider range of error distributions, and a lower learning exponent are not new and confirm previously reported results by already several works in the field.

Abstract: "unprecedented time and system size scales" this phrase appears all too often in MLFF literature, but in the case of this work it is not substantiated. The following sentence mentions "up to hundreds of atoms", while in this field millions of atoms scaling has already been demonstrated by several works.

Table 1: the third column is misleading. For some models L_{max} listed is an architectural constraint (Schnet $L=0$, Pains $L=1$), while for others it's an adjustable hyperparameter. This should be reflected.

P3 "Existing equivariant MLFFs with comparable prediction accuracy would run more than a month for such an analysis." This statement must be accompanied by careful benchmarking comparisons. Fig 1 b is not informative and not substantiated, and likely misleading, since none of the "existing models" are quantified on these axes.

"However, this incomplete list of features can not discriminate certain interaction patterns". This is misleading, this statement should not refer to deep neural networks. Reference 48 demonstrates that local 3-body descriptors do not form a complete representation. But in MPNNs, higher-body correlations are constructed with multiple layers. A different work [<https://iopscience.iop.org/article/10.1088/2632-2153/aca1f8>] demonstrated that distances in MPNNs are not complete, but no such proof exists for angles and dihedral angles as invariants in this context.

"by and attention function" -> "by an attention function"

P5: Li_3SO_4 should be Li_3PO_4 ?

Again, to the point of comparing models, the statement "Remarkably, SO_3 krates achieves energy and force accuracies, more than 50% better than the ones reported in [53]" should not be made without a careful comparison, as it may be misinterpreted as saying one model is capable of higher accuracy than the other. In this particular comparison, in ref 53 the test was performed with a small model to demonstrate scalability to millions of atoms, and presumably not optimized for accuracy. However, the "on par" speed comparison is not systematically supported by sufficient tests of even implementation details (this is just one example in light of my general comments above).

Architecturally, the decoupling of the invariant and equivariant architectures has been previously proposed in the Allegro ref 53. The manuscript should mention how that work relates to the proposed idea.

eSCN [arXiv:2302.03655] reduces the complexity of $\text{SO}(3)$ tensor products from L_{max}^6 to L_{max}^3 , using equivalent $\text{SO}(2)$ convolutions, without avoiding any part of the full operations, as is seemingly done in this manuscript. This should be discussed.

Why do some of the fastest published models, e.g. TorchMD-NET, ANI and others, not appear in Figure 4? The footnote explanation is not easy to understand. It is not appropriate to compare the proposed model to models from two years ago in this field. It is not clear where speed values for the plot for different models come from – were they performed with the same framework on the same GPU, etc? See comments above.

It is not clear to what purpose the authors chose to compare against SGDML in Table 6. This model is no longer state of the art in accuracy and learning efficiency, by more than two years. The result that equivariant models outperform SGDML is well documented.

The proposal of "incorporating atomic cluster expansions into the MP step" was first introduced in <https://arxiv.org/abs/2205.06643>, which should be cited.

Reviewer #1

In this paper by Frank et al the authors present the new architecture “SO3krates” to perform better machine learning construction of potential energy surfaces. In particular they aim at demonstrating that they can improve over known machine learned force fields by performing stable molecular dynamics (MD) simulations for a longer time and by investigating larger size systems.

The paper is well written and addresses an important topic in theoretical physics and chemistry. I think that most of data reported are quite impressive and, therefore, the paper is certainly of potential interest for the readership of Nature Communications.

We thank the reviewer for taking the time to carefully read the manuscript and the positive characterization of our work.

However there is one aspect I think the authors should better clarify and demonstrate. My concern is about what they call the “velocity auto-correlation functions”. On this point I think a revision is necessary for the following reasons:

#1 1) What the authors call “velocity auto-correlation function” is actually the Fourier transform of the velocity auto-correlation function. This should be corrected also in the caption of Figure 6.

The term we used previously was indeed not entirely accurate. We have corrected the wording in the revised manuscript.

#2 2) The authors perform molecular dynamics simulations for a total time of 1ns. Looking at the frequencies of vibrations there is no need for such a long time, unless this is the time need for the thermalization step. Is that the reason of such a long evolution?

We agree that the presented frequency spectra will converge on shorter time scales than the ones performed for our paper. However, one of our goals with this experiment was to demonstrate that MD simulations with the proposed SO3krates MLFF remain stable (and computationally feasible) over extended timescales. We remark that stable, nanosecond-long MD simulations that yield accurate dynamical observables, while visiting regions of configuration space that are not represented in the reference data, represent a significant challenge for current MLFFs due to the potential for error accumulation over time. As such the presented applications go much beyond the current state of the art.

We highlight this aspect more clearly in our revised manuscript within the Results section.

Furthermore, we have included an additional numerical experiment in which we compute the radius of gyration (ROG) for alanine tetrapeptide, which converges much more slowly with time (~100ns in our experiments). The need for MD simulations on the order of tens or even hundreds of nanoseconds to converge this observable is well known in literature (e.g. [1]).

Given the timings measured according to response 1.1a we can perform this simulation within 3 days of wall time whereas other equivariant networks can take up to 27 days for the same simulation. We have added the experiment as well as the figure below to the revised version of our manuscript in the Result section “Radius of Gyration” and in Figure 6.

[1] Yamamoto, Eiji, et al. "Universal relation between instantaneous diffusivity and radius of gyration of proteins in aqueous solution." *Physical review letters* 126.12 (2021): 128101.

#3 3) Investigated temperatures are not high enough. What the authors should is to perform NVE simulations at energy equal to the zero-point energy of the system. Can they perform such a simulation or does the structure break down because they are investigating regions far away from the training region? If they can perform the simulation, what are the frequency of vibrations? They should find some degree of anharmonicity. If so, then they have strongly demonstrated that their new architecture is really able to reach a whole new level of accuracy. If not, it means their PES is basically harmonic and this should be carefully considered by the authors (there is no need of a very refined approach to have a harmonic PES).

We thank the reviewer for the suggestion. To better showcase the learned anharmonicity of our method, we performed an additional experiment on the vibrational modes in the global minimum.

To showcase the anharmonicity of the spectra obtained with SO3krates, we first identify the global minimum of DHA (using the minima hopping results from section F in the revised manuscript). For the found conformation, we calculate harmonic frequencies as well as the zero point energy (ZPE) from harmonic approximation. The ZPE is 12.979 eV, which corresponds to a temperature of roughly 930 K. We find that our method yields stable dynamics despite the large kinetic energy, even though the corresponding temperature is almost twice as high as the one used for generating the training data (500 K). The emergence of non-trivial shifts between the spectra from the NVE (blue curve) with the harmonic frequencies (dashed red lines) allows to illustrate the anharmonicity that our proposed method is able to model. We also investigated the reviewer's hypothesis that simulation temperatures

of 500 K are too low to observe anharmonic effects. We find that, while it is correct that simulations at 500 K show a slightly less pronounced deviation from harmonic behavior, the vibrational spectrum shows no qualitative differences compared to that computed from MD simulations initialized with a kinetic energy corresponding to the ZPE (see Figure below dark blue vs. light blue curve).

We have added the new figure shown below, as well as a description of the new experiment to the revised version of our manuscript (Figure 7 and section F. Power Spectra).

My overall assessment of the paper is certainly positive, but I guess that the authors should properly address the 3 points above before the paper is suitable for acceptance in Nature Communications.

Reviewer #2

The paper “From Peptides to Nanostructures: A Euclidean Transformer for Fast and Stable Machine Learned Force Fields” discusses the development of SO3krates, a transformer architecture that combines the benefit of equivariant representations that usually lead to higher stability in ML-driven molecular dynamics, with the computational efficiency of less accurate representations. The way this is achieved is by replacing the SO(3) convolutions and avoiding expensive tensor products by using a filter on the relative orientation of atomic neighborhoods. The projection applied reduces the representation in a way that only most relevant invariant components of the equivariant basis functions are included that correspond to partial traces of the product-tensor. The forces are obtained as derivatives of invariant energy models. Self-attention layers are used to decouple invariant and equivariant features within the model, which decreases computational complexity compared to other equivariant models. A speed up of about 25 could be achieved in molecular dynamics simulations of small peptides to nanostructures compared to state-of-the-art equivariant models like Nequip, Pains or Allegro without sacrificing accuracy. Finally, the model can be used to detect minimum energy conformations using minima hopping, showcasing the potential for extrapolation, which is especially relevant when scaling up to even larger system. I find the manuscript well written, easy to follow and a major contribution to the field. I only have very little comments specified below.

We thank the reviewer for a thorough evaluation of our manuscript and for acknowledging it as a “major contribution to the field”.

#1 1) It is not fully clear to me whether vectorial and tensorial properties can be directly predicted with SO(3)krates. If the authors could specify this in the text, it would make it more clear.

We thank the reviewer for highlighting this important aspect, which we agree was not described clearly enough, and confirm that SO3krates can also predict vectorial and tensorial properties. This is now clarified at the end of Results section A.

For example, SO3krates is able to provide predictions for equivariant atomic dipoles (using a combination of its $l = 0$ and $l=1$ features) [1] similar to the approach presented in ref. [2]. In the same manner, higher-order tensorial properties, like quadrupoles or octopoles, can be predicted based on $l > 1$ features ($l \geq 1$ features are referred to as "Euclidean variables" in our manuscript).

[1] Unke, Oliver T., et al. "SpookyNet: Learning force fields with electronic degrees of freedom and nonlocal effects." *Nature communications* 12.1 (2021): 7273.

[2] Schütt, Kristof, Oliver Unke, and Michael Gastegger. "Equivariant message passing for the prediction of tensorial properties and molecular spectra." *International Conference on Machine Learning*. PMLR, 2021.

#2 2) Fig. 4c, it would be a nice comparison to see how the RDF looks for invariant models. This would give a better estimate of how close the RDFs really are when comparing ab initio and SO(3)krates.

We thank the reviewer for the suggestion to contrast the distribution of pairwise distances along a MD trajectory obtained with our equivariant SO3krates model and a comparable invariant model.

As we have outlined in our manuscript, one of the key benefits of our equivariant SO3krates model is its exceptional robustness, which brings stability improvements that enable MD simulations on extended timescales. This is particularly important when only limited amounts of reference data are available for training.

To showcase this, we have trained a restricted invariant instance of our model, which we refer to as SO3krates-I0. We remark that this invariant model is representative of the broader field of invariant MLFFs that have been published by the community over the years. In this particular numerical experiment, we find that for 800 data points, both the invariant and the equivariant model generate the correct dynamics compared to the DFT reference data. However, when repeating the experiment with a reduced number of only 200 training data points, the invariant model fails to reproduce the correct distribution of pairwise distances, due to unphysical behavior predicted by the invariant model during the simulation. In contrast, the equivariant model correctly reproduces the distribution of the DFT reference data in this critical low-data condition (see figure below). The figure and a discussion was added to the ms.

#3 3) MD stability: Do the authors mean that they define an MD to be stable when there is a controlled dissociation? What is meant by uncontrolled dissociation?

We agree with the reviewer that the terms ‘controlled’ and ‘uncontrolled’ were potentially confusing. These terms have been replaced them ‘physical’ and ‘non-physical’, respectively, in the revised manuscript. The term ‘non-physical dissociation’ refers to a divergence of the predicted forces/velocities during a dynamics simulation, which results in an instantaneous dissociation of all/some atoms in the molecule. This behavior is clearly unphysical. Such effects can arise from non-smooth potential energy surface models, bad extrapolation behavior towards unseen conformations or error accumulation over subsequent MD steps. In contrast, we use the term ‘physical dissociation’ to describe the effect of physically meaningful dissociation happening on plausible time and kinetic energy scales (see also the Methods section in our revised manuscript, where we have added similar explanations for clarity).

#4 4) Can the authors comment on the computational costs of training SO(3)krates compared to other equivariant models? I think that this information would be very helpful in addition to the speed up in prediction time.

The total training cost can strongly vary depending on the choice of hyperparameters, such as batch size or the number of epochs until convergence of the loss, which itself depends on the number of data points, the learning rate and the optimizer that is used. To make a comparison as fair as possible it thus makes sense to report the time per gradient update.

In ref. [1] (referenced in our manuscript), the total training time has been compared for different state-of-the-art models. Here, e.g. PaiNN takes ~10ms, VISNet takes ~50ms and NequIP or Allegro take ~200ms for Chignolin (166 atoms). Training on the same dataset, we achieve a time per gradient update of ~8ms (measured on an A100GPU) with our model. However, the training times reported in ref. [1] are measured on a V100GPU, which is slower by a factor of 1.6 - 3.4 than our hardware (depending on implementation details). Thus, this runtime comparison should be taken with a grain of salt, but it

illustrates that we can achieve an order of magnitude faster training speed compared to state-of-the-art models such as NequIP or Allegro, while being more accurate with SO3krates.

We remark that the time for each training step depends on the inference speed of the model as well as the number of parameters in the model (due to the need to take the model gradient with respect to all parameters). As highlighted in our manuscript, SO3krates outperforms other equivariant models that are competitive in stability and accuracy in terms of inference speed. Further, SO3krates has only 311k parameters which is significantly less than other equivariant models such as Allegro or NequIP.

These points speak to the fact that our measured training runtimes are in line with our theoretical runtime considerations. We have added a discussion about the training time consumption to the Methods section in our revised manuscript.

[1] Wang, Yusong, et al. "Enhancing geometric representations for molecules with equivariant vector-scalar interactive message passing." *Nature Communications* 15.1 (2024): 313.

#5 5) Extrapolation to larger systems. In such systems, long-range interactions are often not properly captured within the ML representations such that they need to be encountered in non-equivariant models in a "physically-inspired" way, such as it is done in SpookyNet, PhysNet, or Behlers 3rd and 4th generation of NNs or externally via charge predictions (Hirshfeld volumes) as done, e.g., in DOI:10.1039/D2DD00016D or DOI:10.1103/PhysRevB.104.054106. The latter two papers present methods that are also used for materials and hybrid systems including studies on minima searches using basin hopping algorithms. Can the authors comment on the usage of their method for materials? Do they expect that some additional physics in the model is required or do they expect that their representation captures long-range effects properly?

The SO3krates MLFF supports periodic boundary conditions (PBCs), allowing it to be applied to predict properties of materials without any modifications. We now highlight this fact more clearly in our revised manuscript.

To demonstrate the capability of SO3krates to model periodic materials, we have performed MD simulations for Li3PO4 and have calculated the RDF as reported in appendix Fig 15. The training data consists of 11k data points which are randomly sampled from the melted and quenched phase. The starting point for MD simulation is a conformation in the quenched phase and the simulation is run at 600K with a time step of 2fs for a total of 50ps.

Furthermore, we apply our SO3krates model to the selection of materials investigated in ref. [1]. Our prediction accuracies are competitive with the BIGDML model presented in that article, despite not being specifically tailored to the respective symmetry classes of each material (see table below).

		Pd1MgO	Pd500K	Pd1000K	PdH	Na	Graphene
BIGDML	Energy (meV/atom) Forces (meV/Ang)	0.9 41.2	0.1 7.1	0.3 21.2	0.3 17.9	0.3 1.2	0.04 7.1
SO3krates	Energy (meV/atom) Forces (meV/Ang)	0.8 27.7	2.4 14.2	4.1 27.1	0.6 17.5	0.2 1.1	0.07 10.0

Table 1: Energy and force errors for the materials investigated in [1] for SO3krates and BIGDML. The numbers for BIGDML are taken from the original publication [1].

Although equivariant models can improve the extrapolation capabilities for local interactions, this does generally not hold for long-range interactions that go beyond the length scales present in the training data or beyond the effective cutoff of the model. To accurately describe long-range effects when extrapolating to larger structures, methods endowed with the appropriate physical biases are required. Such methods usually rely on the prediction of local properties such as partial charges [2, 3, 4, 5], electronegativities [6] or Hirshfeld volumes [7, 8] they can be seamlessly integrated into our method by learning a

corresponding local descriptor in the invariant feature branch of the SO3krates architecture. We have added a discussion about extrapolation under long-range effects as well as the corresponding references to the Discussion section in the revised version of our manuscript.

We also performed an additional experiment to further analyze the extrapolation capabilities to larger structures. Therefore, we trained a model on 1k training points of Ac-Ala3-NHMe and applied it to study the dynamics of Ala15, starting from the extended structure (very left, figure below). An analysis of the end-to-end distance between the carbonyl carbon of the first residue and the last residue (green spheres) reveals that the peptide rapidly folds into the secondary, helical structure (very right, figure below). A comparison to the end-to-end distance in Ac-Ala3-NHMe (dashed horizontal line) reveals the generalization capabilities of SO3krates towards conformational changes on length-scales that go beyond the ones present in the training data. We have added the experiment as well as the figure from below to section F of the revised version of our manuscript.

- [1] Saucedo, Huziel E., et al. "BIGDML—Towards accurate quantum machine learning force fields for materials." *Nature communications* 13.1 (2022): 3733.
- [2] Unke, Oliver T., et al. "SpookyNet: Learning force fields with electronic degrees of freedom and nonlocal effects." *Nature communications* 12.1 (2021): 7273.
- [3] Unke, Oliver T., and Markus Meuwly. "PhysNet: A neural network for predicting energies, forces, dipole moments, and partial charges." *Journal of chemical theory and computation* 15.6 (2019): 3678-3693.
- [4] Artrith, Nongnuch, Tobias Morawietz, and Jörg Behler. "High-dimensional neural-network potentials for multicomponent systems: Applications to zinc oxide." *Physical Review B* 83.15 (2011): 153101.
- [5] Morawietz, Tobias, Vikas Sharma, and Jörg Behler. "A neural network potential-energy surface for the water dimer based on environment-dependent atomic energies and charges." *The Journal of chemical physics* 136.6 (2012).
- [6] Ko, Tsz Wai, et al. "A fourth-generation high-dimensional neural network potential with accurate electrostatics including non-local charge transfer." *Nature communications* 12.1 (2021): 398.
- [7] Muhli, Heikki, et al. "Machine learning force fields based on local parametrization of dispersion interactions: Application to the phase diagram of C 60." *Physical Review B* 104.5 (2021): 054106.

[8] Westermayr, Julia, et al. "Long-range dispersion-inclusive machine learning potentials for structure search and optimization of hybrid organic–inorganic interfaces." *Digital Discovery* 1.4 (2022): 463-475.

#6 Typo: page 4, right paragraph, "...by and attention function..."

We have corrected this typo in the revised manuscript.

Reviewer #4

Reviewer #4 expressed no methodological or technical concerns, but voiced reservations regarding the performance analysis of our proposed MLFF model. For a better overview, we summarize these concerns as follows (detailed responses to each raised point can be found below):

The precision of our runtime / accuracy measurements is potentially compromised by

- variations in the sophistication of our model's software implementation / hardware utilization compared to the measurements provided in the related works that we quote (points #1, #4, #6, #9, #10),
- differences in the hardware environment (point #13) in the quoted numerical experiments and our own benchmarks.

Furthermore,

- the numbers cited from the original publications of the reference models may not be directly comparable, as these models might not have reached their full potential due to inadequate hyperparameter optimization (#2, partly #9).

They also request further clarifications of some aspects of our manuscript (points #3, #5 - #8, #11 #12, #14, #15).

This work proposes an equivariant machine learning force field (MLFF) model to address the computational speed and accuracy tradeoff. Results include applications to small molecule dynamics and conformation sampling. The main idea is to sparsify the equivariant tensor product operation by avoiding the full product operation and operating only on its invariant output, thereby reducing the scaling with the rotation degree of spherical harmonics.

Overall, the work is interesting, but its impact is limited for several reasons. Proposed architectural modifications to the equivariant neural network architecture are incremental improvements, resembling previous works even if not identically implemented. Moreover, significant advances in "unprecedented" speed and accuracy are claimed but are not systematically demonstrated against state-of-the-art models. Furthermore, only small molecules are considered, and no applications to large more relevant systems are demonstrated, which is the main goal of MLFFs. I do not find this paper appropriate for Nature Communications, but could be a good contribution to a more specialized journal after major revisions. This is a rapidly moving competitive field, and it is important to avoid claims about model speed, stability and accuracy, unless optimal frontiers for each model architecture are considered and compared fairly. Each model has hyperparameters that affect such tradeoffs in different ways. Therefore, a claim of a significantly better model must be substantiated in a systematic quantitative manner.

We politely disagree with the view of the referee that our manuscript is not appropriate for Nature Communications. The methodological advances presented in our manuscript are not incremental and have far-reaching implications for the field of computational chemistry and physics. Our technical contribution (Euclidean attention mechanism) significantly improves the applicability of the currently prevailing modeling paradigm (equivariant SO(3) convolutions) used by modern MLFFs. We firmly believe that the broad set of presented applications and the obtained insights, enabled by our novel developments, make our work appealing to the interdisciplinary readership of Nature Communications and represent an important step for future developments by the community.

#1 Timing tests, especially on small systems, are known to be highly variable and dependent on the ability of NN frameworks to achieve GPU utilization. In addition, the integration of the models into MD codes can also. This paper does not specify which framework was used, which JIT optimizations were used, how the models were implemented with MD codes, how the timings were measured. The overheads of MD codes may or may not be neglected.

We agree with the reviewer that runtime measurements, in particular on GPU, depend on a variety of factors that have to be investigated carefully. However, we want to emphasize that our presented method reduces the theoretical computational complexity with respect to other MLFF models, which is independent of hardware and implementation framework (see Table 1). This is supported by our experimental findings (specifically, MLFF model performances in MD simulations), which show that current state-of-the-art architectures employing SO3 convolutions are the most expensive.

To address the reviewers' concerns about potential discrepancies in implementation, hardware and MD framework, we have re-implemented several representative MLFF models in order to perform comparisons on an equal footing.

Our results are shown in subfigure (a) (below) and they indeed follow the independent measurements presented in ref. [1]. The fastest model (ForceNet) yields wrong observables from the MD trajectories [1], such that we chose the second fastest model (SchNet) for our re-implementation. As the most stable and accurate model we chose NequIP as the second architecture for re-implementation. The chosen architectures are also representative for invariant (SchNet) and equivariant SO3 convolutions (NequIP) constituting the upper and lower bounds in terms of computational complexity (Table 1 in the manuscript). All models are implemented with the same JAX and Python version 3.9. MD step times are measured with the same MD code written in `jax_md` [2] on the same physical device (Nvidia V100 GPU). The models follow the default MD17 hyperparameters as outlined in the original publications [3, 4]. This ensures equal benchmarking conditions for our runtime comparisons that follow. Interestingly, the transition from `pyTorch+ASE` (subfigure (b) purple dashed line) to `JAX+jax_md` (subfigure (b) purple solid line) allows for a speed-up between 28 for NequIP and 15 for SchNet. This illustrates the importance of identical settings and the potential of the JAX ecosystem.

We find that SO3krates can reconcile the somewhat opposing demands of MD stability, speed and accuracy. Our model enables reliable and accurate simulations at sub-microsecond speed (subfigure (b) below). For small organic molecules with up to 21 atoms it achieves an averaged speed-up by a factor of 5 compared to the NequIP architecture whereas step times are slightly larger (by a factor of 1.4) than for the invariant SchNet model. The speed-up over SO3 convolutions increases in the total number of atoms, which is in line with the smaller pre-factor in the theoretical scaling analysis (Table 1 in the manuscript) such that for the double walled nanotube (370 atoms), the speed-up compared to NequIP has grown to a factor of 30 (subfigure (c) below). Compared to invariant convolutions, we find our approach to yield slightly slower prediction speed which is in line with theoretical considerations.

We have added the above description to Section B. Overcoming Accuracy-Stability-Speed Trade-Offs and the new figures from below to Figure 4 in the revised version of our manuscript.

- [1] Fu, Xiang, et al. "Forces are not Enough: Benchmark and Critical Evaluation for Machine Learning Force Fields with Molecular Simulations." *Transactions on Machine Learning Research* (2023).
- [2] Schoenholz, Samuel, and Ekin Dogus Cubuk. "Jax md: a framework for differentiable physics." *Advances in Neural Information Processing Systems* 33 (2020): 11428-11441.
- [3] Schütt, Kristof T., et al. "SchNet—a deep learning architecture for molecules and materials." *The Journal of Chemical Physics* 148.24 (2018).
- [4] Batzner, Simon, et al. "E (3)-equivariant graph neural networks for data-efficient and accurate interatomic potentials." *Nature communications* 13.1 (2022): 2453.

	Ethanol	Salicylic acid	Naphthalene	Aspirin	AcAla3NHMe	DHA	Stachyose	AT-AT-CG-CG	buckyball	nanotube
# atoms	9	16	18	21	42	56	87	118	148	370
SchNet	0.556	0.612	0.598	0.586	0.972	0.857	1.14	1.28	1.64	2.82
NequIP	3.34	4.59	4.54	5.45	10.9	13.6	24.5	27.9	34.8	130.
SO3krates	0.771	0.773	0.801	0.895	1.17	1.25	1.45	1.55	1.66	2.98

Table 2: Runtime comparison for different structures from the MD17 and MD22 dataset for SchNet, SO3krates and NequIP. Reported runtimes are in microseconds.

#2 Without these details and without systematic sweeps through model hyperparameters to establish accuracy-speed pareto frontiers. At this time in the MLFF field, it is unacceptable to publish a paper with claims of model superiority without exploring these considerations.

Hyperparameter searches play a crucial role in deep learning, being a fundamental aspect of model development. Consequently, identifying appropriate hyperparameter ranges constitutes a significant portion of the model development process, and it is safe to assume that this aspect is thoroughly explored by the original authors of the respective MLFF models that we compare to. In our comparisons, we always compare to the recommended “default settings” that are typically provided as part of the definition for each model (e.g. one example is the maximum degree in equivariant MPNNs). We base all our comparisons on the default settings from the respective publications. All performance numbers that we compare our model to, including Fig 4a and Fig 4b are therefore taken from the original publications of the respective models to make sure that no models are disadvantaged by poor implementations / hyperparameter search.

#3 The findings that disabling equivariance leads to less stable models, due to worse extrapolation, wider range of error distributions, and a lower learning exponent are not new and confirm previously reported results by already several works in the field.

Although there has been long standing speculation about the increased stability of equivariant models (see refs. [1, 2, 3] that we cite, and ref. [4], which we have added to the revised manuscript), we provide a comprehensive examination of this hypothesis in our work, which has been missing to date. We would like to emphasize that the simulation timescales explored in our work are beyond reach with existing MLFF models for systems in the size and accuracy regimes presented in our work.

We specifically investigate the following aspects in our manuscript:

- The stability of our proposed MLFF across a wide range of configuration space (see Section II.D, II.E: long-timescale MD simulations and minima searches for large and flexible molecular systems)
- The extrapolation ability of our MLFF model (see Section II.E: accessing new geometries that are not present in the training data)

[1] Fu, Xiang, et al. "Forces are not Enough: Benchmark and Critical Evaluation for Machine Learning Force Fields with Molecular Simulations." *Transactions on Machine Learning Research* (2023).

[2] Stocker, Sina, et al. "How robust are modern graph neural network potentials in long and hot molecular dynamics simulations?." *Machine Learning: Science and Technology* 3.4 (2022): 045010.

[3] Miksch, April M., et al. "Strategies for the construction of machine-learning potentials for accurate and efficient atomic-scale simulations." *Machine Learning: Science and Technology* 2.3 (2021): 031001.

[4] Wang, Zun, et al. "Improving machine learning force fields for molecular dynamics simulations with fine-grained force metrics." *The Journal of chemical physics* 159.3 (2023).

#4 Abstract: “unprecedented time and system size scales” this phrase appears all too often in MLFF literature, but in the case of this work it is not substantiated. The following sentence mentions “up to hundreds of atoms”, while in this field millions of atoms scaling has already been demonstrated by several works.

Our approach represents a step towards reconciling the conflicting demands of long-timescale stability, accuracy and computation efficiency. We remark that our presented approach does not improve along these dimensions unilaterally (which is trivial), but simultaneously. We politely insist that this represents an unprecedented advance in the field, enabling us to run stable 100ns-long MD simulations for flexible systems.

While existing MLFFs have predicted energies and forces for significantly larger systems, such as ALLEGRO, these models have not undergone thorough scrutiny in terms of stability and their capacity for extrapolation. Our own findings and many other works (e.g. [1, 2, 3]) cited in our manuscript and [4] that we have added to the revised version of our manuscript) show that the lack of stability due to error accumulation can be a major applicability roadblock.

However, we agree that there is a need to clarify this circumstance. To improve the presentation of our manuscript, we have therefore changed the corresponding part of our abstract in the revised manuscript.

- [1] Fu, Xiang, et al. "Forces are not Enough: Benchmark and Critical Evaluation for Machine Learning Force Fields with Molecular Simulations." *Transactions on Machine Learning Research* (2023).
- [2] Stocker, Sina, et al. "How robust are modern graph neural network potentials in long and hot molecular dynamics simulations?." *Machine Learning: Science and Technology* 3.4 (2022): 045010.
- [3] Miksch, April M., et al. "Strategies for the construction of machine-learning potentials for accurate and efficient atomic-scale simulations." *Machine Learning: Science and Technology* 2.3 (2021): 031001.
- [4] Wang, Zun, et al. "Improving machine learning force fields for molecular dynamics simulations with fine-grained force metrics." *The Journal of chemical physics* 159.3 (2023).

#5 Table 1: the third column is misleading. For some models L_{max} listed is an architectural constraint (Schnet $L=0$, Pannn $L=1$), while for others it's an adjustable hyperparameter. This should be reflected.

We thank the reviewer for this remark. We have reworked the caption of table 1 in the revised manuscript.

#6 P3 "Existing equivariant MLFFs with comparable prediction accuracy would run more than a month for such an analysis." This statement must be accompanied by careful benchmarking comparisons.

We agree that this must be investigated more carefully. According to the measured runtimes from response **#1.1a**, the achieved speedup for DHA is 12.5x, which results in approximately 25 days compared to 2 days.

#7 Fig 1 b is not informative and not substantiated, and likely misleading, since none of the "existing models" are quantified on these axes.

Fig. 1b is a stylized version of Fig. 4a, which illustrates a fully quantitative benchmarking of the shown MLFFs. To improve the presentation of our manuscript, we have removed Fig. 1b and now exclusively refer to the unabbreviated Fig. 4a in our revised manuscript. We remark that prior studies have independently quantified the stability [1, 2, 3], the speed [4, 5] and both aspects jointly [5].

- [1] Vita, Joshua A., and Daniel Schwalbe-Koda. "Data efficiency and extrapolation trends in neural network interatomic potentials." *Machine Learning: Science and Technology*, 4.3 (2023), 035031.
- [2] Stocker, Sina, et al. "How robust are modern graph neural network potentials in long and hot molecular dynamics simulations?." *Machine Learning: Science and Technology* 3.4 (2022): 045010.
- [3] Wang, Zun, et al. "Improving machine learning force fields for molecular dynamics simulations with fine-grained force metrics." *The Journal of chemical physics* 159.3 (2023).
- [4] Wang, Yusong, et al. "Enhancing geometric representations for molecules with equivariant vector-scalar interactive message passing." *Nature Communications* 15.1 (2024): 313.
- [5] Fu, Xiang, et al. "Forces are not Enough: Benchmark and Critical Evaluation for Machine Learning Force Fields with Molecular Simulations." *Transactions on Machine Learning Research* (2023).

#8 "However, this incomplete list of features can not discriminate certain interaction patterns". This is misleading, this statement should not refer to deep neural networks. Reference 48 demonstrates that local 3-body descriptors do not form a complete representation. But in MPNNs, higher-body correlations are constructed with multiple layers. A different work [<https://iopscience.iop.org/article/10.1088/2632-2153/aca1f8>] demonstrated that distances in MPNNs are not complete, but no such proof exists for angles and dihedral angles as invariants in this context.

We thank the reviewer for pointing this out and we fully agree. We have removed the statement in the revised manuscript and have added the corresponding references.

- "by and attention function" -> "by an attention function"

- P5: Li3SO4 should be Li3PO4?

We have corrected the typos in the revised manuscript.

#9 Again, to the point of comparing models, the statement “Remarkably, SO3krates achieves energy and force accuracies, more than 50% better than the ones reported in [53]” should not be made without a careful comparison, as it may be misinterpreted as saying one model is capable of higher accuracy than the other. In this particular comparison, in ref 53 the test was performed with a small model to demonstrate scalability to millions of atoms, and presumably not optimized for accuracy.

We have clarified that the model referenced in [53] was optimized for speed rather than accuracy. The performance of our model is comparable to that model, yet our model can achieve a significantly higher accuracy (about 50% better) given the same speed constraint.

#10 However, the “on par” speed comparison is not systematically supported by sufficient tests of even implementation details (this is just one example in light of my general comments above).

For our comparison we reference the runtimes from the original ALLEGRO paper, which we assume to have been optimized to best demonstrate the model's potential.

#11 Architecturally, the decoupling of the invariant and equivariant architectures has been previously proposed in the Allegro ref 53. The manuscript should mention how that work relates to the proposed idea.

To maintain invariance of the potential energy surface prediction, all models treat invariant and equivariant features separately. However, the innovation of our approach lies in the a priori separation of both interaction spaces. This enables us to avoid tensor products, which are a key component of all existing equivariant networks to date, including the Allegro architecture. Further, the Allegro architecture works on edge based representations whereas our method is centered around atomic representations, which allows for improvements in computational time and space complexity. We have added a discussion about the relation to separated interaction spaces as well as to the Allegro architecture to the revised version of our manuscript.

#12 eSCN [arXiv:2302.03655] reduces the complexity of SO(3) tensor products from L_{\max}^6 to L_{\max}^3 , using equivalent SO(2) convolutions, without avoiding any part of the full operations, as is seemingly done in this manuscript. This should be discussed.

We thank the reviewer for making us aware of this interesting approach. The referenced method simplifies SO3 convolutions, whereas our method allows us to fully avoid expensive SO3 convolutions within the message passing paradigm. Both works show orthogonal directions to improve upon the current state-of-the-art when representing equivariant interaction in MLFFs, indicating that this active field of research has not yet fully matured. We have added an appropriate discussion about this to the revised manuscript to the Discussion section.

#13 Why do some of the fastest published models, e.g. TorchMD-NET, ANI and others, not appear in Figure 4? The footnote explanation is not easy to understand. It is not appropriate to compare the proposed model to models from two years ago in this field. It is not clear where speed values for the plot for different models come from – were they performed with the same framework on the same GPU, etc? See comments above.

We compare our model to results that have been published in an extensive study evaluating 8 representative MPNN-based MLFFs in an equal setting [1]. The chosen architectures range from invariant

convolutional neural networks (SchNet), ones that use vector representations (PaiNN), to fully SO(3)-equivariant models (NequIP), thus covering the full evolution of MPNN based MLFFs.

This comparison includes recently proposed models (e.g. PaiNN, ForceNet from 2021, NequIP from 2022), as well as some older models (e.g. SchNet from 2017) that remain competitive in some aspects, such as computational efficiency. We firmly believe that it is of scholarly importance to consider relevant architectures that have been published before 2022. In that spectrum, PaiNN has been chosen as the representative architecture for MPNNs with vectorial representations, the class that TorchMD-NET also belongs to. Since PaiNN and TorchMD-NET lie in the same complexity class, their computational performance only differs by a factor (between 1.1 for the large variant of TorchMD-NET and 1.4 for the smaller variant of TorchMD-NET as shown in the original publication [1]).

To address any potentially remaining concerns, we have re-implemented a selection of MLFF that cover the spectrum of modeling approaches (see our response **#1.1.a**). A detailed description of these runtime experiments have been added to the revised manuscript including an updated caption in Fig. 4.

[1] Thölke, Philipp, and Gianni De Fabritiis. "Torchmd-net: equivariant transformers for neural network based molecular potentials." *arXiv preprint arXiv:2202.02541* (2022).

#14 It is not clear to what purpose the authors chose to compare against SGDML in Table 6. This model is no longer state of the art in accuracy and learning efficiency, by more than two years. The result that equivariant models outperform SGDML is well documented.

We report comparisons to sGDML in table 3, since it is the paper which introduced the benchmark dataset. We further want to stress that the sheer error numbers do not tell much about a model's usefulness for practical applications [1, 2] and the table is only intended to showcase that chemical accuracy can be achieved.

[1] Fu, Xiang, et al. "Forces are not Enough: Benchmark and Critical Evaluation for Machine Learning Force Fields with Molecular Simulations." *Transactions on Machine Learning Research* (2023).
[2] Stark, Wojciech G., et al. "Machine Learning Interatomic Potentials for Reactive Hydrogen Dynamics at Metal Surfaces Based on Iterative Refinement of Reaction Probabilities." *The Journal of Physical Chemistry C* 127.50 (2023): 24168-24182.

#15 The proposal of "incorporating atomic cluster expansions into the MP step" was first introduced in <https://arxiv.org/abs/2205.06643>, which should be cited.

We have added the citation to the revised version of our manuscript.

REVIEWERS' COMMENTS

Reviewer #1 (Remarks to the Author):

The authors have addressed all my comments properly. Therefore I am happy to suggest this manuscript for publication.

Reviewer #2 (Remarks to the Author):

The authors have addressed all my concerns and I am happy with the current version.